# Adjustable Robust Reinforcement Learning for Online 3D Bin Packing

**Yuxin Pan**[1]    **Yize Chen**[2*]    **Fangzhen Lin**[3*]

[1]EMIA, The Hong Kong University of Science and Technology
[2]AI Thrust, The Hong Kong University of Science and Technology (Guangzhou)
[3]CSE, The Hong Kong University of Science and Technology
`yuxin.pan@connect.ust.hk`   `yizechen@ust.hk`   `flin@cs.ust.hk`

## Abstract

Designing effective policies for the online 3D bin packing problem (3D-BPP) has been a long-standing challenge, primarily due to the unpredictable nature of incoming box sequences and stringent physical constraints. While current deep reinforcement learning (DRL) methods for online 3D-BPP have shown promising results in optimizing average performance over an underlying box sequence distribution, they often fail in real-world settings where some worst-case scenarios can materialize. Standard robust DRL algorithms tend to overly prioritize optimizing the worst-case performance at the expense of performance under normal problem instance distribution. To address these issues, we first introduce a permutation-based attacker to investigate the practical robustness of both DRL-based and heuristic methods proposed for solving online 3D-BPP. Then, we propose an adjustable robust reinforcement learning (AR2L) framework that allows efficient adjustment of robustness weights to achieve the desired balance of the policy's performance in average and worst-case environments. Specifically, we formulate the objective function as a weighted sum of expected and worst-case returns, and derive the lower performance bound by relating to the return under a mixture dynamics. To realize this lower bound, we adopt an iterative procedure that searches for the associated mixture dynamics and improves the corresponding policy. We integrate this procedure into two popular robust adversarial algorithms to develop the exact and approximate AR2L algorithms. Experiments demonstrate that AR2L is versatile in the sense that it improves policy robustness while maintaining an acceptable level of performance for the nominal case.

## 1   Introduction

The offline 3D bin packing problem (3D-BPP) is a classic NP-hard combinatorial optimization problem (COP) (de Castro Silva et al., 2003), which aims to optimally assign cuboid-shaped items with varying sizes to the minimum number of containers while satisfying physical constraints (Martello et al., 2000). In such a setting, a range of physical constraints can be imposed to meet diverse packing requirements and preferences (Gzara et al., 2020). The primary constraint requires items to be packed stably, without any overlaps, and kept within the bin. The online counterpart of 3D-BPP is more prevalent and practical in logistics and warehousing (Wang and Hauser, 2020), as it does not require complete information about all the unpacked items in advance. In this setting, only a limited number of upcoming items on a conveyor can be observed, and items must be packed after the preceding items are allocated (Seiden, 2002). Thus, besides physical constraints already present

---

[*]Co-corresponding Authors

in the offline counterpart, the item permutation on the conveyor introduces additional constraints, while the solution approach must take packing order into consideration.

Solution techniques for online 3D-BPP can be broadly categorized into heuristic and learning-based methods. As heuristics often heavily rely on manually designed task-specific score functions (Chazelle, 1983; Ha et al., 2017), they have limitations in expressing complex packing preferences and adapting to diverse scenarios. By comparison, learning-based methods usually involve utilizing deep reinforcement learning (DRL) techniques to develop a more adaptable packing policy capable of accommodating diverse packing requirements (Zhao et al., 2021, 2022a; Song et al., 2023). While emerging DRL-based methods for online 3D-BPP are effective at optimizing average performance, they do not account for worst-case scenarios. This is because these methods often fail on "hard" box sequences arising from the inherent uncertainty in incoming box sequences. In addition, few studies have investigated the robustness of these methods, as incoming item sequences can vary a lot in practice. These limit the practical usage of learning-based approaches.

To study the algorithm robustness under worst-case scenarios, one plausible approach is to design an attacker that can produce perturbations commonly encountered in real-world settings. Various proposed attackers for perturbing RL models often add continuous-value noise to either received observations or performed actions (Tessler et al., 2019; Sun et al., 2021). Yet such methods are mostly not well-suited for online 3D-BPP, as the state is defined by the bin configuration and the currently observed items, and adding continuous-value noise may not correspond to real-world perturbations. As previously discussed, item permutations can significantly impact the performance of a given packing strategy, and in real-world scenarios, the packing strategy may face challenging box sequences. By contrast, perturbing the bin configuration may violate practical constraints or packing preferences. Therefore, we argue that *using diverse item permutations to evaluate the model's robustness* is a more suitable approach. Some of general robust reinforcement learning frameworks against perturbations can be applied to online 3D-BPP (Pinto et al., 2017; Ying et al., 2022; Ho et al., 2022; Panaganti et al., 2022), yet these methods may develop overly conservative policies, as they usually prioritize improving worst-case performance at the expense of average performance. This motivates us to design a robust algorithm that can effectively handle worst-case problem instances while attaining satisfactory average performance.

In this paper, we develop a novel permutation-based attacker to practically evaluate robustness of both DRL-based and heuristic bin packing policies. We propose an Adjustable Robust Reinforcement Learning (AR2L) framework, which preserves high performance in both nominal and worst-case environments. Our learnable attacker selects an item and place it to the most preceding position in the observed incoming item sequence, as only the first item in this sequence has the priority to be packed first. Essentially, this attacker attempts to identify a problem instance distribution that is challenging for a given packing policy by simply reordering item sequences. The developed AR2L incorporates such novel attacker along with an adjustable robustness weight parameter. Specifically, to consider both the average and worst-case performance, we formulate the packing objective as a weighted sum of expected and worst-case returns defined over space utilization rate. We derive a lower bound for the objective function by relating it to the return under a mixture dynamics, which guarantees AR2L's performance. To realize this lower bound, we turn to identifying *a policy with the highest lower bound as the surrogate task*. In this task, we use an iterative procedure that searches for the associated mixture dynamics for the policy evaluation, and improves the policy under the resulting mixture dynamics. To further put AR2L into practice, we find the connecting link between AR2L with the Robust Adversarial Reinforcement Learning (RARL) algorithm (Pinto et al., 2017) and the Robust $f$-Divergence Markov Decision Process (RfMDP) algorithm (Ho et al., 2022; Panaganti et al., 2022), resulting in the exact AR2L and the approximate AR2L algorithms respectively. Empirical evidence shows our method is able to achieve competitive worst-case performance in terms of space utilization under the worst-case attack, while still maintaining good average performance compared to all previously proposed robust counterparts.

## 2  Related Work

Numerous heuristic Ha et al. (2017); Karabulut and İnceoğlu (2005); Wang and Hauser (2019); Li and Zhang (2015); Hu et al. (2017, 2020) and learning-based methods (Hu et al., 2017; Verma et al., 2020; Zhao et al., 2021, 2022b; Yang et al., 2021; Zhao et al., 2022a; Song et al., 2023) have been proposed to solve online 3D-BPP. However, these methods typically only consider average performance and

may not be robust against perturbations. For more detailed reviews of related algorithms on 3D-BPP, please refer to the Appendix.

**Adversarial Attacks in Deep RL.** More recently, several studies have demonstrated that deep RL algorithms are vulnerable to adversarial perturbations from attackers, and robust policies can be trained accordingly (Huang et al., 2017; Zhang et al., 2020, 2021; Sun et al., 2021; Ding et al., 2023). Yet standard DRL attack schemes cannot generate realistic attacks for 3D-BPP due to the setting of added perturbations. In the field of COP, Lu et al. (2023) used a RL-based attacker to modify the underlying graph. Yet this method is limited to offline COPs that can be formulated as graph. Kong et al. (2019) employed a generative adversarial network (GAN) (Goodfellow et al., 2020) to generate some worst-case problem instances from Gaussian noise under restricted problem setting.

**Robust RL.** In order to find policies that are robust against various types of perturbations, a great number of studies have investigated different strategies, such as regularized-based methods (Zhang et al., 2020; Shen et al., 2020; Oikarinen et al., 2021; Kumar et al., 2021; Kuang et al., 2022), attack-driven methods (Kos and Song, 2017; Pattanaik et al., 2017; Behzadan and Munir, 2017), novel bellman operators (Liang et al., 2022; Wang and Zou, 2021, 2022), and conditional value at risk (CVaR) based methods (Chow et al., 2017; Tang et al., 2019; Hiraoka et al., 2019; Ying et al., 2022). However, these methods are generally intended to deal with $l_p$-norm perturbations on vectorized observations, which could limit their applicability to online 3D-BPP. Built upon robust MDP framework (Iyengar, 2005), Pinto et al. (2017) modeled the competition between the agent and the adversary as a zero-sum two-player game, but such game formulation has an excessive prioritization of the worst-case performance. Jiang et al. (2021) introduced Monotonic Robust Policy Optimization (MRPO) to enhance domain generalization by connecting worst-case and average performance. Yet such approach imposes Lipschitz continuity assumptions, which are unaligned with 3D-BPP. The recently developed robust $f$-divergence MDP can approximate the worst-case value of a policy by utilizing nominal samples instead of relying on samples from an adversary (Ho et al., 2022; Panaganti et al., 2022). However, this method still cannot account for expected cases, as its focus is solely on learning values in worst-case scenarios.

## 3 Preliminaries

**MDP Formulation of Online 3D-BPP.** To learn a highly effective policy via RL, the online 3D-BPP is formulated as an MDP. Inspired by PCT (Zhao et al., 2022a), in this formulation, the state $s_t^{\text{pack}} \in \mathcal{S}^{\text{pack}}$ observed by the packing agent consists of the already packed $N_C$ items $\mathbf{C}_t$ in the bin, the observed incoming $N_B$ items $\mathbf{B}_t$, and a set of potential positions $\mathbf{L}_t$. The packed items in $\mathbf{C}_t$ have spatial properties like sizes and coordinates, while each item $b_{t,i} \in \mathbf{B}_t, i \in \{1, .., N_B\}$ only provides size information. The potential positions are typically generated for the most preceding item in $\mathbf{B}_t$ using heuristic methods (Martello et al., 2000; Crainic et al., 2008; Ha et al., 2017). Then, the action $a_t^{\text{pack}} \in \mathcal{A}^{\text{pack}}$ is to choose one position $l_{t,j} \in \mathbf{L}_t, j \in \{1, .., N_L\}$ for the first item in $\mathbf{B}_t$. In the termination state $s_T$ ($T$ is the episode length), the agent cannot pack any more items. As a result, the agent receives a delayed reward $r_T$ that represents the space utilization at $s_T$, instead of immediate rewards ($r_t = 0, t < T$). The discount factor $\gamma$ here is set to 1. The aim is to maximize the space utilization by learning a stochastic packing policy $\pi_{\text{pack}}(l_{t,j}|\mathbf{C}_t, \mathbf{B}_t, \mathbf{L}_t)$.

**Robust MDP.** The goal of robust MDP is to find the optimal robust policy that maximizes the value against the worst-case dynamics from an uncertainty set $\mathcal{P}^w$. The uncertainty set is defined in the neighborhood of the nominal dynamics $P^o = (P_{s,a}^o, (s, a) \in \mathcal{S} \times \mathcal{A})$ and satisfies rectangularity condition (Iyengar, 2005), defined as $\mathcal{P}^w = \otimes_{(s,a) \in \mathcal{S} \times \mathcal{A}} \mathcal{P}_{s,a}^w$, $\mathcal{P}_{s,a}^w = \{P_{s,a} \in \Delta(\mathcal{S}) : D_{TV}(P_{s,a} || P_{s,a}^o) \leq \rho\}$, where $D_{TV}(\cdot)$ denotes the total variation (TV) distance, $\otimes$ is the Cartesian product, $\Delta(\mathcal{S})$ is a set of probability measures defined on $\mathcal{S}$, and $\rho \in [0, 1]$ is the radius of the uncertainty set. Consequently, the robust value function under a policy $\pi$ is $V_r^\pi = \inf_{P^w \in \mathcal{P}^w} V_r^{\pi, P^w}$. And the robust Bellman operator $\mathcal{T}_r$ is defined as $\mathcal{T}_r V_r^\pi(s) = \mathbb{E}_{a \sim \pi}[r(s, a) + \gamma \inf_{P_{s,a}^w \in \mathcal{P}_{s,a}^w} \mathbb{E}_{s' \sim P_{s,a}^w}[V_r^\pi(s')]]$, where $s'$ denotes the next state. To empirically solve the robust MDP, Pinto et al. (2017) proposed RARL that learns the robust policy under the environment perturbed by a learned optimal adversary.

**Robust $f$-divergence MDP.** To avoid the need for a costly trained adversary, RfMDP (Ho et al., 2022; Panaganti et al., 2022) formulates the problem as a tractable constrained optimization task to approximate the robust value using samples from $P^o$. The objective is to find a transition distribution

that minimizes the robust value function, subject to the constraint in $\mathcal{P}^w$ described by the $f$-divergence. As a result, a new robust Bellman operator is introduced by solving the dual form of this constrained optimization problem through the Lagrangian multiplier method.

# 4 Adjustable Robust Reinforcement Learning

In this section, we begin by introducing a novel yet practical adversary capable of generating worst-case problem instances for the online 3D-BPP. Next, we present the adjustable robust reinforcement learning (AR2L) framework to address the robustness-performance tradeoff. Finally, we integrate AR2L into both the RARL algorithm and the RfMDP algorithm to derive the exact and approximate AR2L algorithms in a tractable manner.

## 4.1 Permutation-based Attacker

In online 3D-BPP, a problem instance is comprised of a sequence of items, and these items are randomly permuted and placed on a conveyor. Such randomness can result in certain instances where trained policy may fail. Therefore, this observation motivates us to design a simple yet realistic adversary called permutation-based attacker. By reordering the item sequence for a given packing policy, our approach can explore worst-case instances without compromising realism (See Figure 1 for attacker overview). In contrast to the approach

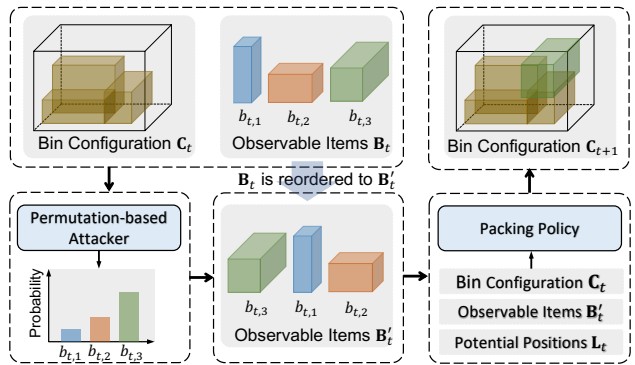

Figure 1: Overview of our attack framework.

of directly permuting the entire item sequence as described in (Kong et al., 2019), our method involves permuting the observable item sequence, thereby progressively transforming the entire item sequence. Through the behavior analysis of our permutation-based attacker in the Appendix, we observe the attacker appears to prefer smaller items when constructing harder instances as the number of observable items increases. This aligns with the findings mentioned in (Zhao et al., 2022a), where larger items simplify the scenario while smaller items introduce additional challenges. As a result, the act of permuting the entire sequence might enable the attacker to trickily select certain types of items, and thus carries potential risks associated with shifts in the underlying item distribution. However, we aim to ensure that the attacker genuinely learns how to permute the item sequence to identify and create challenging scenarios. To mitigate these risks, we limit the attacker's capacity by restricting the number $N_B$ of observable items to proactively address concerns related to changes in the item distribution.

To target any type of solver intended for solving the online 3D-BPP for robustness evaluation, we train a RL-based policy which acts as the permutation-based attacker. The permuted state $s_t^{\mathrm{perm}} \in \mathcal{S}^{\mathrm{perm}}$ of the attacker is comprised of the current bin configuration $\mathbf{C}_t$ and the item sequence $\mathbf{B}_t$. The action $a_t^{\mathrm{perm}} \in \mathcal{A}^{\mathrm{perm}}$ involves moving one item from $\mathbf{B}_t$ to a position ahead of the other observed items, resulting in the reordered item sequence $\mathbf{B}_t' = \{b_{t,i}'\}$. Then, the packing policy will pack the most preceding one into the bin based on the perturbed observation. The reward for the attacker is defined as $r_t^{\mathrm{perm}} = -r_t^{\mathrm{pack}}$, since the objective is to minimize the space utilization by training an optimal stochastic policy $\pi_{\mathrm{perm}}(b_{t,i}|\mathbf{C}_t, \mathbf{B}_t)$. The online packing process allows only the first item in the observable item sequence to be packed at each step. To address the issue of exponential growth in the action space as the number of observable items increases, the permutation-based attacker strategically selects one item and positions it ahead of the other items. We can consider a simplified value function represented as $V^{\pi_{\mathrm{pack}}}(\mathbf{C}_t, \mathbf{B}_t') = r_t^{\mathrm{pack}} + V^{\pi_{\mathrm{pack}}}(\mathbf{C}_{t+1}, \mathbf{B}_{t+1}')$. This function indicates that the value of $\pi_{\mathrm{pack}}$ depends on $r_t^{\mathrm{pack}}$, $\mathbf{C}_{t+1}$, and $\mathbf{B}_{t+1}'$. Furthermore, $r_t^{\mathrm{pack}}$ and $\mathbf{C}_{t+1}$ are influenced by the first item in $\mathbf{B}_t'$, while the remaining items in $\mathbf{B}_t'$ combine with a new item to form a new sequence, which undergoes further permutations as $\mathbf{B}_{t+1}'$. As a result, the permutation of the remaining items at time step $t$ is disregarded in the attack process. To model such

an attacker, we use the Transformer (Vaswani et al., 2017) that is capable of capturing long-range dependencies in spatial data and sequential data involved in the attacker's observation. Please refer to the Appendix for more details on the implementations.

## 4.2 Adjustable Robust Reinforcement Learning

For online 3D-BPP, item observed at each timestep is independently generated from a stationary distribution $p_b(b_{t,i})$ that does not change over time. Once an item is placed into the container, the bin configuration $\mathbf{C}_t$ becomes deterministic, and a new item $b_{t+1,N_B}$ is appended to the observable item sequence to construct $\mathbf{B}_{t+1}$. Thus, the nominal environment is defined as $P^o(\mathbf{C}_{t+1}, \mathbf{B}_{t+1}|\mathbf{C}_t, \mathbf{B}_t, a_t^{\text{pack}}) = p_b(b_{t+1,N_B})$. We omit $\mathbf{L}_t$ for brevity. Since we use the permutation-based attacker to reorder the observable item sequence, the worst-case environment transition is $P^w(\mathbf{C}_{t+1}, \mathbf{B}'_{t+1}|\mathbf{C}_t, \mathbf{B}'_t, a_t^{\text{pack}}) = p_b(b_{t+1,N_B})\pi_{\text{perm}}(b'_{t+1,1}|\mathbf{C}_{t+1}, \mathbf{B}_{t+1})$, with $P^w$ as the worst-case dynamics.

Existing DRL-based methods are developed to learn a packing policy that maximizes space utilization (cumulative reward) under the nominal distribution $P^o$. However, in real-world scenarios where the order of items can be adversarially permuted, it is not sufficient to only consider the expected return under $P^o$. In contrast, the existing general robust methods that can be deployed to online 3D-BPP overly prioritize robustness at the cost of average performance by solely focusing on the return under the worst-case dynamics $P^w$. To address these issues, we should be aware of the returns under both the average and worst-case scenarios. Given that nominal cases are more common than worst-case scenarios in the online 3D-BPP setting, the objective function is defined as

$$\pi^* = \arg\max_{\pi \in \Pi} \eta(\pi, P^o) + \alpha\eta(\pi, P^w) \tag{1}$$

where $\alpha \in (0, 1]$ is the weight of robustness, $\eta(\cdot)$ is the return, and $P^w$ is built on the optimal permutation-based attacker. Here symbols without superscripts represent those used by the packing agent (e.g., $\pi_{\text{pack}} = \pi$).

However, how can we learn such a policy with the presence of both $P^o$ and $P^w$? To address this matter, we derive a lower bound for the objective function defined in Equation 1 by relating it to the return under an unknown mixture dynamics defined as $P^m$, as shown in the following theorem.

**Theorem 1.** *The model discrepancy between two models can be described as $d(P^1||P^2) \triangleq \max_{s,a} D_{TV}(P^1(\cdot|s,a)||P^2(\cdot|s,a))$. The lower bound for objective 1 is derived as follows:*

$$\eta(\pi, P^o) + \alpha\eta(\pi, P^w) \geq (1+\alpha)\eta(\pi, P^m) - \frac{2\gamma|r|_{\max}}{(1-\gamma)^2}(d(P^m||P^o) + \alpha d(P^m||P^w)) \tag{2}$$

The RHS of Inequality 2 provides the lower bound for the objective function defined in Equation 1, where the first term represents the expected return of policy $\pi$ under the mixture dynamics $P^m$, while the second term denotes the weighted sum of deviations of $P^m$ from both $P^o$ and $P^w$. Motivated by Theorem 1, we turn to maximizing the RHS of Inequality 2 to concurrently improve both the average and worst-case performance. Therefore, the primal problem in Equation 1 is equivalently reformulated to a surrogate problem where we expect to identify an optimal policy with the maximal lower bound, represented as

$$\pi^* = \arg\max_{\pi \in \Pi} \max_{P^m \in \mathcal{P}} \eta(\pi, P^m) - (d(P^m||P^o) + \alpha d(P^m||P^w)); \tag{3}$$

The objective of this optimization problem is to maximize the return by updating both the policy and the mixture dynamics controlled by robustness weight $\alpha$, given the nominal and worst-case dynamics. Furthermore, the second term in Equation 3 can be viewed as a constraint that penalizes the deviations between environments, resulting in a constrained optimization problem written as $\pi^* = \arg\max_{\pi \in \Pi}\{\max_{P^m \in \mathcal{P}} \eta(\pi, P^m) : d(P^m||P^o) + \alpha d(P^m||P^w) \leq \rho'\}$, where $\rho' \in [0, 1+\alpha]$. However, this constraint renders the constrained form of Equation 3 impractical to solve. Instead, a heuristic approximation is used for this constraint, which considers the TV distance at each state-action pair, leading to an uncertainty set for the mixture dynamics:

$$\mathcal{P}^m = \otimes_{(s,a)\in\mathcal{S}\times\mathcal{A}}\mathcal{P}^m_{s,a}; \ \mathcal{P}^m_{s,a} = \{P_{s,a} \in \Delta(\mathcal{S}) : D_{TV}(P_{s,a}||P^o_{s,a}) + \alpha D_{TV}(P_{s,a}||P^w_{s,a}) \leq \rho'\}. \tag{4}$$

The uncertainty set $\mathcal{P}^m$ satisfies the rectangularity condition Iyengar (2005). To solve the constrained optimization problem, we carry out an iterative procedure which searches for the associated mixture

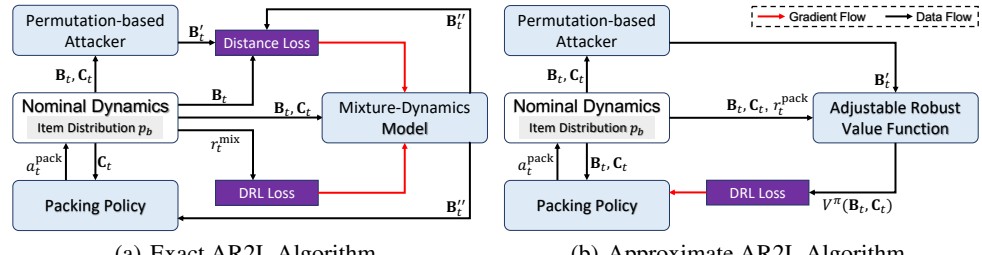

(a) Exact AR2L Algorithm       (b) Approximate AR2L Algorithm

Figure 2: Implementations of the AR2L algorithm. **Left:** The exact AR2L algorithm requires to learn a mixture-dynamics model to generate problem instances for the training of the packing policy. **Right:** The approximate AR2L algorithm relies on the samples from both the nominal dynamics and the permutation-based attacker to estimate adjustable robust values of the packing policy.

dynamics for the policy evaluation, and improves the policy under the resulting mixture dynamics. Therefore, we define the adjustable robust value function as $V_a^\pi = \sup_{P^m \in \mathcal{P}^m} V_a^{\pi, P^m}$ for the policy evaluation. And the adjustable robust Bellman operator $\mathcal{T}_a$ can be defined as

$$\mathcal{T}_a V_a^\pi(s) = \mathbb{E}_{a \sim \pi}[r(s, a) + \gamma \sup_{P_{s,a}^m \in \mathcal{P}_{s,a}^m} \mathbb{E}_{s' \sim P_{s,a}^m}[V_a^\pi(s')]]. \tag{5}$$

The adjustable robust Bellman operator assumes both the nominal and worst-case dynamics corresponding to $\pi$ are available for constructing the uncertainty set $\mathcal{P}^m$. Unlike the traditional robust Bellman operator (Iyengar, 2005) designed for the minimal value, our proposed operator focus on maximizing the value of a given policy by updating the mixture dynamics. Meanwhile, to avoid the optimistic evaluation, the uncertainty set for the mixture dynamics used in Equation 13 is constrained with regard to both nominal and worst-case environment. Following policy evaluation, the policy is improved using samples drawn from the mixture dynamics. In addition, the following theorem guarantees that $\mathcal{T}_a$ can facilitate the convergence of the value function to a fixed point.

**Theorem 2.** *For any given policy $\pi$ and its corresponding worst-case dynamics $P^w$, the adjustable robust Bellman operator $\mathcal{T}_a$ is a contraction whose fixed point is $V_a^\pi$. The operator equation $\mathcal{T}_a V_a^\pi = V_a^\pi$ has a unique solution.*

**Exact AR2L Algorithm.** The AR2L algorithm evaluates and improves policies based on the mixture dynamics that exists in the neighborhoods of both the nominal and worst-case scenarios. However, this algorithm remains impractical since the adjustable robust Bellman operator defined in Equation 13 involves computing the expectations w.r.t. all models in the uncertainty set $\mathcal{P}^m$. This potentially results in high computational complexity. Inspired by the classic RARL algorithm (Pinto et al., 2017) which trains a robust policy utilizing perturbations injected by a learned optimal adversary, we propose an iterative training approach that involves training a mixture-dynamics model $\pi_{\text{mix}}$ alongside the packing policy $\pi_{\text{pack}}$ and the permutation-based attacker $\pi_{\text{perm}}$ to obtain the exact AR2L algorithm. The mixture-dynamics model is corresponding to the mixture dynamics, and it permutes the observable item sequences $\mathbf{B}_t$ to $\mathbf{B}_t''$ based on observed packed items $\mathbf{C}_t$. As illustrated in Figure 2(a), such a model receives the same reward as the packing policy ($r_t^{\text{mix}} = r_t^{\text{pack}}$). Based on Equation 3, the mixture-dynamics model strives to maximize the return under a given policy, while also penalizing deviations between dynamics. As $P^m$ and $P^w$ have the same form with the same underlying item distribution $p_b$, we can only measure the discrepancy between $P^m$ and $P^w$ by evaluating the deviation between $\pi_{\text{mix}}$ and $\pi_{\text{perm}}$. In practice, the Kullback-Leibler (KL) divergence is adopted to constrain the deviation between distributions. Thus, the training loss is

$$\mathcal{L}_{\text{mix}} = -\eta(\pi_{\text{pack}}, \pi_{\text{mix}}) + (D_{KL}(\pi_{\text{mix}} || \mathbb{1}_{\{x=b_{t+1,1}\}}) + \alpha D_{KL}(\pi_{\text{mix}} || \pi_{\text{perm}})), \tag{6}$$

where $D_{KL}$ is the KL divergence, and $\mathbb{1}_{\{x=b_{t+1,1}\}}$ denotes the indicator function. Thus, the mixture-dynamics model is optimized by minimizing both the DRL loss (the first term) and the distance loss (the second term). Based on this, the exact AR2L algorithm iterates through three stages to progressively improve the packing policy. First, the permutation-based attacker is optimized for a given packing policy, as is done in RARL. Next, the mixture-dynamics model $\pi_{\text{mix}}$ is learned using the loss function defined in Equation 6. Finally, the packing policy is evaluated and improved using problem instances that are permuted by $\pi_{\text{mix}}$.

**Approximate AR2L Algorithm.** Training the mixture-dynamics model in exact AR2L introduces additional computation to the entire framework. We thus propose the approximate AR2L algorithm, which uses samples from both the nominal and worst-case dynamics to estimate adjustable robust values for the policy iteration as shown in Figure 2(b). Inspired by RfMDP (Ho et al., 2022; Panaganti et al., 2022), we use the dual reformulation of the adjustable robust Bellman operator, as given below.

**Proposition 1.** *The policy evaluation of AR2L can be formulated as a constrained optimization problem. The objective is to maximize the value function over the uncertainty set $\mathcal{P}^m$, subject to the constraint defined in Equation 4. By representing the TV distance using the $f$-divergence (Shapiro, 2017), the adjustable robust Bellman operator $\mathcal{T}_a$ given in Equation 13 can be equivalently written as*

$$\mathcal{T}_a V_a^\pi(s) = \mathbb{E}_{a\sim\pi}[r(s,a) + \frac{\gamma}{1+\alpha} \inf_{\lambda,\mu_1,\mu_2,\mu} (\mathbb{E}_{s'\sim P_{s,a}^o}[V_a^\pi(s') - \mu_1(s')]_+ + \\ \alpha\mathbb{E}_{s'\sim P_{s,a}^w}[V_a^\pi(s') - \mu_2(s')]_+ + \mu + \lambda\rho(1+\alpha))], \tag{7}$$

*where $[x]_+$ denotes $\max\{x, 0\}$, $\mu_1(s)$ and $\mu_2(s)$ are Lagrangian multipliers for each $s \in \mathcal{S}$; $\mu = \mu_1(s) + \alpha\mu_2(s)$ holds for each $s \in \mathcal{S}$, and $\lambda = \max_s\{V_a^\pi(s) - \mu_1(s), V_a^\pi(s) - \mu_2(s), 0\}$.*

The approximate AR2L algorithm eliminates the requirement of using samples from the mixture dynamics for policy evaluation, yet it introduces estimation errors into the process. In practice, this results in mild performance degradation and sometimes unstable learning. Nevertheless, it remains valuable as the approximate AR2L still demonstrates superior performance compared to RfMDP.

In order to implement the AR2L algorithm, we opt to use the PPO algorithm (Schulman et al., 2017) to train packing policy $\pi_{\text{pack}}$, permutation-based attacker $\pi_{\text{perm}}$ and mixture-dynamics model $\pi_{\text{mix}}$. The packing policy is implemented using the PCT method (Zhao et al., 2022a) to accommodate the continuous solution space. All of the relevant implementation details, pseudocode for the algorithms, and both derivations and proofs are included in the Appendix.

## 5 Experiments

**Training and Evaluation Settings** In the online 3D-BPP setting, the container sizes $S^d, d \in \{x, y, z\}$ are equal for each dimension ($S^x = S^y = S^z$), and the item sizes $s^d, d \in \{x, y, z\}$ are limited to no greater than $S^d/2$ to create more complex scenarios. The stability of each item is checked based on constraints used in (Zhao et al., 2022a,b). Moreover, we adopt two different popular settings used in (Zhao et al., 2022a,b) for training DRL-based policies. In the discrete setting, where item sizes are uniformly generated from a discrete set, i.e., $s^d \in \{1, 2, 3, 4, 5\}$, resulting in a total of 125 types of items, and the container size $S^d$ is fixed at 10. In the continuous setting, the container size $S^d$ is set to 1, and item sizes $s^d, d \in \{x, y\}$ follow a continuous uniform distribution $U(0.1, 0.5)$; and $s^z$ is uniformly sampled from a finite set $\{0.1, 0.2, 0.3, 0.4, 0.5\}$. Then, discrete and continuous datasets are created following instructions from the two aforementioned training settings to evaluate both heuristic and DRL-based methods. Each dataset consists of 3,000 problem instances, where each problem instance contains 150 items. We use three metrics to evaluate the performance of various packing strategies: $Uti.$ represents the average space utilization rate of the bin's volume; $Std.$ is the standard deviation of space utilization which evaluates the algorithm's reliability across all instances; $Num.$ evaluates the average number of packed items.

### 5.1 Algorithm Robustness under Permutation-based Attacks

In our experiment, we evaluate the robustness of six heuristic methods, including deep-bottom-left (DBL) method (Karabulut and İnceoğlu, 2005), best-match-first (BMF) method (Li and Zhang, 2015), least-surface-area heuristics (LSAH) (Hu et al., 2017), online bin packing heuristics (OnlineBPH) (Ha et al., 2017), heightmap-minimization (HMM) method (Wang and Hauser, 2019) and maximize-accessible-convex-space (MACS) method (Hu et al., 2020). For the DRL-based methods, we evaluate the constrained deep reinforcement learning (CDRL) (Zhao et al., 2021) algorithm, and the packing configuration tree (PCT) method (Zhao et al., 2022a). We train specific permutation-based attackers for each of them. Since the capacity of the permutation-based attacker is highly related to the number of observable items of the attacker, we choose $N_B = 5, 10, 15, 20$ to obtain different attackers. Given that the number of observable items for the packing policy is typically limited to the first item in most methods, except for PCT, we can establish a consistent approach by setting the number of observable items for the packing policy to 1 in this case.

Table 1: The performance of existing methods under the perturbation in the discrete setting.

| Methods | $N_B = 5$ | | | $N_B = 10$ | | | $N_B = 15$ | | | $N_B = 20$ | | | w/o attack | | |
|---|---|---|---|---|---|---|---|---|---|---|---|---|---|---|---|
| | *Uti.(%)* | *Std.* | *Num.* | *Uti.(%)* | *Std.* | *Num.* | *Uti.(%)* | *Std.* | *Num.* | *Uti.(%)* | *Std.* | *Num.* | *Uti.(%)* | *Std.* | *Num.* |
| DBL | 40.4 | 14.3 | 18.4 | 28.5 | 16.1 | 13.9 | 26.0 | 14.9 | 14.9 | 21.3 | 12.6 | 8.5 | 63.6 | 11.9 | 25.8 |
| BMF | 46.9 | 11.5 | 21.0 | 40.9 | 12.2 | 22.2 | 38.6 | 12.9 | 21.1 | 33.7 | 11.9 | 22.1 | 62.0 | 9.2 | 24.8 |
| LSAH | 46.0 | 10.1 | 20.3 | 42.2 | 11.3 | 20.9 | 38.6 | 11.0 | 19.4 | 35.6 | 11.8 | 21.3 | 60.9 | 10.9 | 24.6 |
| OnlineBPH | 47.1 | 21.0 | 19.8 | 44.0 | 18.9 | 22.8 | 29.8 | 18.5 | 14.6 | 22.4 | 12.7 | 14.1 | 64.1 | 8.9 | 25.8 |
| HMM | 49.1 | 11.1 | 22.5 | 46.5 | 13.8 | 22.5 | 43.4 | 13.0 | 21.4 | 40.4 | 10.0 | 24.1 | 56.1 | 10.4 | 22.6 |
| MACS | 43.0 | 9.7 | 21.9 | 40.8 | 9.0 | 24.8 | 39.0 | 9.8 | 23.7 | 38.3 | 9.0 | 27.0 | 53.0 | 10.8 | 21.5 |
| CDRL | 61.5 | 7.8 | 27.9 | 56.1 | 6.9 | 28.7 | **54.6** | 7.6 | 31.3 | **51.0** | 7.1 | 29.9 | 74.1 | 7.3 | 29.1 |
| PCT | **63.6** | 9.9 | 27.3 | **58.7** | 11.3 | 25.8 | 50.9 | 13.1 | 25.6 | 40.5 | 15.3 | 21.8 | **76.6** | 6.0 | 30.0 |

Table 1 presents the performance of various methods under perturbations from corresponding permutation-based attackers in the discrete setting. We can observe that though the HMM algorithm does not perform the best in the nominal scenario compared to other heuristic methods, it exhibits superior performance under attackers with different attack capacities. By comparison, the DBL method is the most vulnerable heuristic method. As attack capabilities increase, the CDRL algorithm demonstrates greater robustness compared to the PCT method, consistently achieving a higher average number of packed items with smaller variance of volume ratio. It is worth noting that the larger the value of $N_B$ is, the more the performance can be degraded for all methods, indicating that harder problem instances can be explored by increasing the value of $N_B$. In addition, the number of packed items does not necessarily decrease as attack capacity increases. This is because permutation-based attackers aim to minimize space utilization, which means they may select smaller items to make the packing problem more challenging for the packing policy.

## 5.2 Performance of AR2L Algorithm

We then benchmark four other RL algorithms compatible with online 3D-BPP tasks, which include packing configuration tree (PCT) (Zhao et al., 2022a) method, the CVaR-Proximal-Policy-Optimization (CPPO) (Ying et al., 2022) algorithm, the robust adversarial reinforcement learning (RARL) algorithm (Pinto et al., 2017), and the robust $f$-divergence MDP (RfMDP) (Ho et al., 2022; Panaganti et al., 2022). The PCT algorithm serves as the baseline method and is used as the packing policy for all other methods in our implementations. The CPPO algorithm trains the packing policy with worst-case returns as a constraint, which can potentially improve both average and worst-case performance. The RARL algorithm is the baseline model for the exact AR2L algorithm due to its reliance on worst-case samples from the attacker. Likewise, the RfMDP framework serves as the baseline model for the approximate AR2L algorithm because it involves approximating values of the unknown dynamics. As both the exact AR2L (ExactAR2L) algorithm and the approximate AR2L algorithm (ApproxAR2L) can develop policies with different levels of robustness by adjusting the value of $\alpha$, we choose $\alpha = 0.3, 0.5, 0.7, 1.0$ to explore the relationship between $\alpha$ and policy robustness. Furthermore, we use different attackers with varying attack capabilities ($N_B = 5, 10, 15, 20$) to investigate the robustness of packing policies. It is important to note that in this scenario, the packing policy is permitted to observe the same number of items as its corresponding attacker. During testing, we construct mixture datasets by randomly selecting $\beta\%$ nominal box sequences and reordering them using the learned permutation-based attacker for each packing policy. The empirical results in the continuous setting are included in the Appendix. Since we implement PCT with different training configurations, we conducted a comparison between our implementation and the official implementation in the Appendix.

*Uti.* Table 2 presents the results of evaluating multiple robust methods on various datasets that differ in the number of problem instances perturbed by different attackers. We denote the different values of $\alpha$ used by the ExactAR2L algorithm as ExactAR2L($\alpha$), and similarly for ApproxAR2L($\alpha$). The results unequivocally indicate that the ExactAR2L algorithm, when employed with a higher value of $\alpha$, exhibits superior performance on nominal cases and competitive robustness in terms of space utilization when compared to other robust methods. When the value of $\alpha$ is increased, it allows the packing policy to encounter and learn from more worst-case samples during training. By experiencing a wider range of adversarial situations, the packing policy can better adapt to mitigate the impact of attackers, enhancing its overall robustness. However, ExactAR2L cannot consistently improve its average performance across all the settings with different values of $N_B$. For the scenario

Table 2: The performance of robust methods under the perturbation in the discrete setting.

| $N_B$ | Methods | $\beta=0$ Uti.(%) | Std. | Num. | $\beta=25$ Uti.(%) | Std. | Num. | $\beta=50$ Uti.(%) | Std. | Num. | $\beta=75$ Uti.(%) | Std. | Num. | $\beta=100$ Uti.(%) | Std. | Num. |
|---|---|---|---|---|---|---|---|---|---|---|---|---|---|---|---|---|
| $N_B=5$ | PCT | 76.2 | 6.8 | 29.9 | 73.6 | 7.9 | 29.9 | 70.6 | 9.2 | 28.5 | 67.5 | 9.1 | 27.8 | 64.9 | 8.2 | 27.1 |
| | CPPO | 75.5 | 6.9 | 29.6 | 73.1 | 8.0 | 29.0 | 70.7 | 8.4 | 28.5 | 67.9 | 8.2 | 27.9 | 65.2 | 7.3 | 27.4 |
| | RARL | 74.6 | 6.2 | 29.2 | 73.1 | 7.3 | 29.2 | 70.8 | 8.4 | 29.0 | 67.8 | 8.4 | 28.4 | **66.7** | 8.1 | 28.5 |
| | ExactAR2L(0.3) | 76.3 | 6.6 | 29.8 | 73.8 | 8.5 | 29.6 | 71.0 | 9.1 | 29.2 | 67.5 | 9.6 | 28.4 | 64.5 | 9.4 | 27.7 |
| | ExactAR2L(0.5) | 76.5 | 7.1 | 30.1 | 73.5 | 8.8 | 29.4 | 70.5 | 10.4 | 28.8 | 68.2 | 10.3 | 28.1 | 65.3 | 10.0 | 27.6 |
| | ExactAR2L(0.7) | 76.8 | 6.1 | 30.1 | **74.4** | 7.7 | 29.7 | 71.1 | 8.9 | 29.1 | 68.8 | 9.7 | 28.7 | 66.3 | 8.6 | 28.5 |
| | ExactAR2L(1.0) | **77.4** | 6.5 | 30.2 | 74.3 | 9.3 | 29.5 | **72.0** | 9.5 | 29.0 | **69.5** | 10.0 | 28.4 | 66.5 | 8.7 | 27.9 |
| | RfMDP | 75.5 | 6.7 | 29.6 | 71.8 | 9.8 | 28.8 | 69.6 | 9.4 | 28.5 | 67.1 | 9.0 | 28.0 | 63.9 | 8.1 | 27.3 |
| | ApproxAR2L(0.3) | 75.3 | 6.3 | 29.6 | 72.6 | 7.7 | 28.9 | 70.0 | 8.6 | 28.4 | 66.4 | 9.7 | 27.6 | 63.2 | 9.5 | 26.9 |
| | ApproxAR2L(0.5) | 76.1 | 5.8 | 29.8 | 72.6 | 8.4 | 29.0 | 69.8 | 9.9 | 28.4 | 67.3 | 9.6 | 27.7 | 64.7 | 8.8 | 27.4 |
| | ApproxAR2L(0.7) | 76.9 | 5.7 | 30.0 | 73.7 | 8.0 | 29.1 | 71.6 | 8.5 | 28.7 | 68.2 | 8.4 | 27.8 | 65.2 | 7.9 | 27.0 |
| | ApproxAR2L(1.0) | 76.5 | 6.5 | 30.1 | 73.4 | 8.9 | 29.2 | 70.8 | 9.3 | 28.6 | 68.4 | 9.3 | 28.0 | 65.7 | 8.5 | 27.5 |
| $N_B=10$ | PCT | 76.4 | 6.6 | 29.9 | 70.6 | 12.4 | 28.5 | 65.1 | 14.5 | 27.3 | 61.4 | 14.8 | 26.4 | 55.7 | 12.9 | 25.2 |
| | CPPO | 75.6 | 7.2 | 29.8 | 70.7 | 11.6 | 28.6 | 66.2 | 12.7 | 27.7 | 62.3 | 12.9 | 26.9 | 57.4 | 11.4 | 25.8 |
| | RARL | 74.3 | 7.2 | 29.4 | 71.1 | 8.7 | 29.0 | 69.2 | 8.5 | 28.7 | 65.6 | 8.6 | 28.0 | 63.3 | 8.2 | 27.7 |
| | ExactAR2L(0.3) | 76.5 | 5.9 | 29.9 | 71.5 | 12.1 | 28.9 | 66.8 | 14.0 | 27.8 | 62.9 | 14.1 | 27.0 | 59.2 | 12.7 | 26.4 |
| | ExactAR2L(0.5) | **77.6** | 5.8 | 30.3 | **73.1** | 10.2 | 29.5 | 68.0 | 13.1 | 28.8 | 64.0 | 12.7 | 28.0 | 59.7 | 10.2 | 27.4 |
| | ExactAR2L(0.7) | 77.4 | 5.6 | 30.2 | **73.1** | 9.7 | 29.5 | 69.5 | 11.2 | 28.9 | 65.6 | 11.1 | 28.1 | 62.3 | 9.7 | 27.4 |
| | ExactAR2L(1.0) | 76.0 | 7.0 | 29.8 | 72.4 | 9.7 | 30.0 | **70.3** | 9.4 | 30.3 | **66.7** | 9.3 | 30.3 | **63.8** | 8.0 | 30.6 |
| | RfMDP | 74.4 | 7.2 | 29.7 | 70.5 | 11.4 | 28.7 | 65.7 | 14.3 | 28.0 | 60.8 | 14.4 | 26.8 | 55.9 | 12.5 | 25.9 |
| | ApproxAR2L(0.3) | 76.1 | 5.9 | 29.7 | 70.5 | 12.5 | 28.8 | 66.1 | 14.6 | 28.2 | 61.3 | 14.5 | 27.2 | 55.7 | 11.8 | 26.2 |
| | ApproxAR2L(0.5) | 76.2 | 5.9 | 29.9 | 72.1 | 11.7 | 29.2 | 66.9 | 14.7 | 28.1 | 62.1 | 15.0 | 26.9 | 56.1 | 13.6 | 25.5 |
| | ApproxAR2L(0.7) | 73.0 | 7.0 | 28.8 | 70.1 | 9.5 | 29.2 | 65.4 | 11.1 | 28.4 | 61.0 | 10.7 | 27.9 | 56.2 | 8.4 | 27.1 |
| | ApproxAR2L(1.0) | 73.6 | 6.8 | 28.9 | 69.3 | 10.7 | 29.2 | 66.1 | 11.8 | 29.3 | 61.9 | 11.6 | 29.4 | 57.1 | 9.1 | 29.8 |
| $N_B=15$ | PCT | 76.8 | 6.9 | 29.9 | 69.4 | 15.4 | 28.1 | 62.0 | 18.3 | 26.3 | 55.2 | 18.2 | 24.3 | 48.6 | 14.7 | 22.5 |
| | CPPO | 75.2 | 7.7 | 29.3 | 69.9 | 13.1 | 28.2 | 63.8 | 15.2 | 26.8 | 58.0 | 15.7 | 25.2 | 52.3 | 13.3 | 23.9 |
| | RARL | 73.2 | 7.2 | 28.7 | 70.3 | 9.1 | 28.7 | 67.2 | 10.9 | 28.4 | 63.1 | 12.4 | 28.0 | **58.7** | 11.0 | 27.5 |
| | ExactAR2L(0.3) | 76.8 | 6.4 | 30.1 | 71.4 | 11.4 | 29.3 | 65.3 | 13.8 | 28.4 | 59.7 | 14.0 | 27.3 | 53.4 | 11.6 | 26.5 |
| | ExactAR2L(0.5) | **77.7** | 5.8 | 30.3 | **72.0** | 11.9 | 29.3 | **68.0** | 13.8 | 28.8 | 61.2 | 15.4 | 27.2 | 55.0 | 13.3 | 26.0 |
| | ExactAR2L(0.7) | 76.5 | 7.5 | 30.0 | 71.5 | 13.4 | 29.0 | 66.4 | 14.8 | 28.0 | 62.0 | 14.5 | 26.7 | 56.3 | 12.3 | 25.6 |
| | ExactAR2L(1.0) | 76.6 | 5.4 | 30.0 | 71.7 | 10.8 | 29.4 | 67.7 | 12.3 | 29.0 | **63.3** | 11.9 | 28.2 | 58.5 | 11.1 | 27.6 |
| | RfMDP | 73.6 | 8.0 | 29.0 | 68.4 | 12.8 | 29.1 | 62.2 | 15.0 | 28.9 | 58.2 | 14.7 | 29.2 | 54.1 | 12.5 | 29.6 |
| | ApproxAR2L(0.3) | 75.4 | 7.3 | 29.5 | 69.9 | 12.3 | 29.4 | 65.5 | 13.5 | 29.2 | 59.3 | 13.2 | 28.0 | 54.5 | 10.6 | 27.5 |
| | ApproxAR2L(0.5) | 75.2 | 6.4 | 29.6 | 69.9 | 10.6 | 29.5 | 65.2 | 11.9 | 29.5 | 60.1 | 10.9 | 29.2 | 55.1 | 6.8 | 29.4 |
| | ApproxAR2L(0.7) | 74.6 | 7.5 | 29.2 | 70.2 | 10.3 | 29.1 | 65.0 | 11.9 | 28.8 | 60.6 | 11.5 | 28.6 | 55.8 | 8.0 | 28.6 |
| | ApproxAR2L(1.0) | 73.5 | 7.5 | 29.1 | 69.1 | 10.2 | 29.0 | 65.0 | 12.4 | 29.3 | 60.4 | 11.6 | 29.0 | 55.8 | 8.8 | 29.2 |
| $N_B=20$ | PCT | **77.0** | 5.5 | 30.1 | 68.4 | 17.0 | 28.6 | 59.7 | 19.3 | 27.0 | 50.9 | 18.2 | 25.5 | 41.9 | 12.2 | 24.2 |
| | CPPO | 74.1 | 7.5 | 29.2 | 66.7 | 16.2 | 27.9 | 59.1 | 18.5 | 26.5 | 53.2 | 17.3 | 25.3 | 45.8 | 13.6 | 24.0 |
| | RARL | 72.0 | 6.4 | 28.4 | 68.6 | 9.4 | 29.1 | 64.6 | 10.6 | 29.3 | 61.7 | 10.0 | 30.0 | **58.7** | 8.4 | 30.4 |
| | ExactAR2L(0.3) | 76.7 | 6.7 | 30.0 | 70.8 | 13.3 | 29.4 | 65.6 | 15.8 | 28.9 | 59.4 | 16.3 | 28.0 | 52.7 | 14.3 | 27.4 |
| | ExactAR2L(0.5) | 76.8 | 6.2 | 30.1 | 70.0 | 14.3 | 30.2 | 64.7 | 15.8 | 30.5 | 60.0 | 15.3 | 30.6 | 54.4 | 13.3 | 30.8 |
| | ExactAR2L(0.7) | 76.3 | 6.1 | 30.0 | **71.1** | 11.4 | 29.6 | 66.2 | 14.1 | 29.3 | 61.9 | 14.2 | 29.2 | 57.0 | 12.3 | 28.8 |
| | ExactAR2L(1.0) | 76.1 | 7.3 | 30.0 | 70.9 | 11.8 | 29.4 | **66.7** | 12.7 | 29.0 | **62.8** | 12.6 | 28.6 | 58.5 | 10.3 | 28.2 |
| | RfMDP | 73.8 | 7.0 | 29.0 | 69.4 | 11.0 | 26.6 | 64.7 | 13.3 | 24.2 | 59.4 | 11.7 | 21.5 | 54.4 | 13.0 | 19.2 |
| | ApproxAR2L(0.3) | 76.1 | 6.1 | 29.8 | 70.3 | 12.1 | 30.4 | 64.1 | 14.1 | 30.8 | 57.7 | 12.3 | 31.0 | 51.7 | 6.7 | 31.5 |
| | ApproxAR2L(0.5) | 75.0 | 7.6 | 29.5 | 70.1 | 12.2 | 30.1 | 63.9 | 15.4 | 30.5 | 58.8 | 14.4 | 30.7 | 53.1 | 11.7 | 31.1 |
| | ApproxAR2L(0.7) | 74.1 | 6.9 | 29.0 | 68.9 | 10.8 | 29.3 | 64.0 | 11.9 | 30.0 | 59.4 | 11.3 | 30.4 | 54.1 | 7.1 | 30.5 |
| | ApproxAR2L(1.0) | 73.4 | 8.2 | 28.9 | 68.2 | 13.2 | 28.7 | 65.6 | 13.9 | 29.0 | 61.9 | 14.6 | 28.9 | 57.6 | 12.9 | 28.8 |

where $N_B = 5$, the distribution deviation between the perturbed data and the nominal data ($\beta = 0$) is comparatively minimal. This indicates that increasing the value of $\alpha$ and incorporating more challenging instances into the training settings of the nominal dynamics is an acceptable approach. By doing so, the generalization of the model can be enhanced, allowing it to handle a wider range of scenarios. However, the distribution deviation between the perturbed data and the nominal data ($\beta = 0$) increases as the number of observable items increases. In such cases, it becomes less desirable to significantly increase the value of $\alpha$. As a result, in the cases of $N_B = 10$ and $N_B = 15$ under $\beta = 0$, ExactAR2L tends to favor $\alpha = 0.5$. Furthermore, in the scenario of $N_B = 20$, ExactAR2L

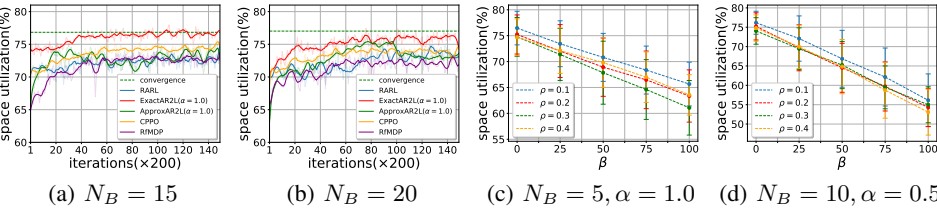

(a) $N_B = 15$  (b) $N_B = 20$  (c) $N_B = 5, \alpha = 1.0$  (d) $N_B = 10, \alpha = 0.5$

Figure 3: 3(a) 3(b) depict the learning curves of robust RL algorithms for nominal problem instances. 3(c) 3(d) show the influence of $\rho$ on the performance of the ApproxAR2L algorithm.

cannot outperform PCT due to the larger distribution deviation caused by the increased number of observable items of the attacker. In a word, by including perturbed data with a small deviation from the nominal data during training, the enhanced generalization leads to the performance improvement in nominal cases ($\beta = 0$). Conversely, the large distribution deviation between the perturbed data and the nominal data can degenerate the performance in nominal cases, as indicated by the RARL algorithm. We also observe that as the value of $\beta$ increases, the ExactAR2L algorithm tends to favor a larger value of $\alpha$. Furthermore, the ApproxAR2L algorithm exhibits a similar performance tendency in both nominal and worst-case scenarios as the ExactAR2L algorithm. Due to the introduced value estimation error, the ApproxAR2L algorithm cannot perform as well as the ExactAR2L algorithm. Despite this, the ApproxAR2L algorithm still performs better than its baseline model, the RfMDP algorithm. Additionally, as demonstrated in Figures 3(a) 3(b), the ExactAR2L algorithm can learn faster than other robust RL algorithms.

$\boldsymbol{Std.}$ As shown in Table 2, ExactAR2L demonstrates its superiority over PCT with smaller $Std.$ in 17 tasks, while producing a slightly larger $Std.$ in 3 tasks. Thus, ExactAR2L can indeed improve the robustness. Furthermore, we observe when $N_B = 5, 10$, ExactAR2L tends to choose $\alpha = 0.7$ for smaller $Std.$. On the other hand, for $N_B = 15, 20$, ExactAR2L favors $\alpha = 1.0$. Since ExactAR2L is trained on both nominal and worst-case dynamics, while RARL is trained only on the worst-case dynamics, the ExactAR2L policy is less conservative than the RARL policy. While the conservativeness may result in smaller $Std.$ in most tasks, it produces worse results in terms of $Uti.$ under the nominal dynamics. It is worth noting that the value of $Std.$ from ExactAR2L is the closest to that of RARL. This observation shows ExactAR2L can trade off between conservative and risky behavior, as the $Std.$ from ExactAR2L is between that of RARL and PCT. Similarly, ApproxAR2L is less conservative than RfMDP, which causes ApproxAR2L cannot achieve a smaller $Std.$ in all tasks.

$\boldsymbol{Num.}$ The ExactAR2L(1.0) algorithm can pack more items in 17 tasks compared to PCT, and shows a slight drop in 3 tasks, where the average drop is 0.2. We found that to pack more items, ExactAR2L consistently favors $\alpha = 1.0$ across various tasks. Compared to RARL, ExactAR2L(1.0) can pack at least the same number of items in 16 tasks. Thus ExactAR2L(1.0) can produce competitive results compared to RARL and PCT in terms of $Num.$. Compared to the baseline method RfMDP, ApproxAR2L(0.5) can pack more items in 16 tasks, and shows a slight drop in only 4 tasks, where the average drop is 0.25.

**Hyperparameter Configurations.** Based on observations from Table 2, $\alpha = 1.0$ is the best choice for ExactAR2L across different test settings. When $\beta = 50, 75$, ExactAR2L(1.0) performs the best compared to baselines and ExactAR2L with other values of $\alpha$ (with a slight drop compared to ExactAR2L(0.7) when $\beta = 50, N_B = 15$). If $\beta = 100$, ExactAR2L(1.0) can still produce competitive results compared to RARL and significantly outperforms PCT. When $\beta = 25$, although $\alpha = 1.0$ is not the optimal choice, ExactAR2L(1.0) can still outperform other baselines. When $\beta = 0$, ExactAR2L(1.0) significantly outperforms RARL, and the slight drop compared to PCT is acceptable, as our goal is to improve the robustness while maintaining average performance at an acceptable level. $\rho$ is only used in ApproxAR2L algorithm. As shown in Figures 3(c) 3(d), we choose different values of $\rho$ in different settings. We found that $\rho = 0.1$ is a trustworthy choice for ApproxAR2L. Based on the observations from Table 2 and Figures 3(c) 3(d), we conclude that $\rho = 0.1$ and $\alpha = 0.5$ are the best choice for ApproxAR2L, as it outperforms its corresponding baseline in almost all the tasks.

## 6 Conclusions and Limitations

In this work, we propose a novel reinforcement learning approach, Adjustable Robust Reinforcement Learning (AR2L), for solving the online 3D Bin Packing Problem (BPP). AR2L achieves a balance between policy performance in nominal and worst-case scenarios by optimizing a weighted sum of returns. We use a surrogate task of identifying a policy with the largest lower bound of the return to optimize the objective function. We can seamlessly bridge AR2L into the RARL and RfMDP algorithms to obtain exact and approximate AR2L algorithms. Extensive experiments show that the exact AR2L algorithm improves the robustness of the packing policy while maintaining acceptable performance, but may introduce additional computational complexity due to the mixture-dynamics model. The approximate AR2L algorithm estimates values without samples from the mixture dynamics, yet performance is not up to our exact AR2L agent due to the existence of estimation error. Though our AR2L framework is designed for online 3D-BPP, it can be also adapted to other decision-making tasks. We will investigate more efficient and generalizable AR2L framework.

## Acknowledgments and Disclosure of Funding

We appreciate the anonymous reviewers, (S)ACs, and PCs of NeurIPS2023 for their insightful comments to further improve our paper and their service to the community. We would like to thank the Turing AI Computing Cloud (TACC) (Xu et al., 2021) and HKUST iSING Lab for providing us computation resources on their platform.

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

# A  Appendix

## A.1  Experiment

In addition to results in the discrete setting presented earlier in the main texts, this section focuses on introducing additional results in the continuous setting. For the continuous setting, the container size $S^d, d \in \{x, y, z\}$ is fixed at 1, and the sizes of items $s^d, d \in \{x, y\}$ follow a continuous uniform distribution $U(0.1, 0.5)$. To prevent impractical scenarios where packed items cannot form a supportive plane for incoming items, the $s^z$ dimension of each item is uniformly sampled from a finite set of values $\{0.1, 0.2, 0.3, 0.4, 0.5\}$ (Zhao et al., 2022a). Likewise, Each dataset consists of 3,000 problem instances, where each problem instance contains 150 items. We use three metrics to evaluate the performance of various packing strategies: $Uti.$ represents the average space utilization rate of the bin's volume; $Std.$ is the standard deviation of space utilization across all instances; $Num.$ evaluates the average number of packed items. **In the field of online 3D-BPP (Zhao et al., 2021, 2022a; Song et al., 2023), the standard deviation of space utilization across a large number of problem instances (>1,000) is commonly used to assess the reliability and stability of a solver, rather than the error bar generated from multiple experiment runs (Kim et al., 2022; Choo et al., 2022; Qiu et al., 2022)**. In addition, all the models are developed using PyTorch (Paszke et al., 2017) and trained on a Nvidia RTX 3090 GPU and an Intel(R) Xeon(R) Gold 5218R CPU @ 2.10GHz.

### A.1.1  Algorithm Robustness under Permutation-based Attacker

In the continuous setting, we conduct empirical evaluations to assess the robustness of four heuristic methods: deep-bottom-left (DBL) method (Karabulut and İnceoğlu, 2005), best-match-first (BMF) method (Li and Zhang, 2015), least-surface-area heuristic (LSAH) (Hu et al., 2017), and online bin packing heuristics (OnlineBPH) (Ha et al., 2017). In addition, we evaluated the packing configuration tree (PCT) algorithm (Zhao et al., 2022a) for the DRL-based methods, as other DRL-based methods are limited to spatially discretized grid world. We also evaluate the robustness of each algorithm under varying attack capabilities by selecting different values for $N_B$, including 5, 10, 15, and 20. Since except PCT, other methods can observe only the first item in the item sequence at each time step, the observable number of items of each packing strategy is fixed at 1 to ensure a fair comparison between these packing strategies.

Table 1 displays the performance of different packing strategies against permutations generated by their corresponding permutation-based attackers. We can see that the LSAH algorithm exhibits superior space utilization performance across various experimental settings with different values of $N_B$, even though it does not perform as well as the OnlineBPH and DBL algorithms in the nominal dynamics. In contrast, the BMF method is the most susceptible to attacks across all settings with varying numbers of observable items used by permutation-based attackers. With the increase of the attack capability, the PCT algorithm consistently outperforms the heuristic methods in terms of space utilization and the number of packed items, which can be attributed to its generalizability across a large number of problem instances. Similar to the results in the discrete setting, increasing the value of $N_B$ results in more performance degradation for all methods, which suggests an opportunity to explore more challenging problem instances, and to design proper robust algorithms accordingly.

Table 1: The performance of existing methods under the perturbation in the continuous setting.

| Methods | $N_B = 5$ | | | $N_B = 10$ | | | $N_B = 15$ | | | $N_B = 20$ | | | w/o attack | | |
|---|---|---|---|---|---|---|---|---|---|---|---|---|---|---|---|
| | $Uti.(\%)$ | $Std.$ | $Num.$ | $Uti.(\%)$ | $Std.$ | $Num.$ | $Uti.(\%)$ | $Std.$ | $Num.$ | $Uti.(\%)$ | $Std.$ | $Num.$ | $Uti.(\%)$ | $Std.$ | $Num.$ |
| DBL | 37.8 | 8.7 | 17.3 | 30.7 | 8.3 | 16.1 | 27.1 | 6.5 | 16.1 | 23.0 | 7.0 | 15.9 | 48.3 | 10.5 | 18.6 |
| BMF | 27.5 | 8.6 | 13.3 | 21.6 | 7.3 | 12.5 | 17.4 | 6.5 | 12.0 | 14.1 | 4.7 | 11.0 | 42.1 | 8.2 | 16.4 |
| LSAH | 38.7 | 10.9 | 16.7 | 36.9 | 9.1 | 17.2 | 33.2 | 10.0 | 16.3 | 27.1 | 8.4 | 17.6 | 47.5 | 8.6 | 18.4 |
| OnlineBPH | 33.8 | 11.6 | 15.3 | 30.2 | 7.4 | 16.5 | 26.4 | 7.2 | 16.2 | 23.3 | 6.8 | 15.6 | 49.1 | 9.7 | 19.0 |
| PCT | **47.7** | 8.4 | 20.4 | **39.0** | 8.7 | 18.5 | **36.4** | 9.5 | 18.8 | **30.4** | 8.8 | 17.9 | **65.6** | 8.7 | 25.0 |

Figure 1 depicts item distributions under perturbations from different attackers using PCT packing policies in the discrete setting. Since the sizes $s^d$ of items follow a uniform discrete distribution, the frequency of the sampled sizes are approximately equal, as illustrated in the leftmost bar chart in Figure 1. It is apparent that as the number of observable items of the attacker increases, the attacker

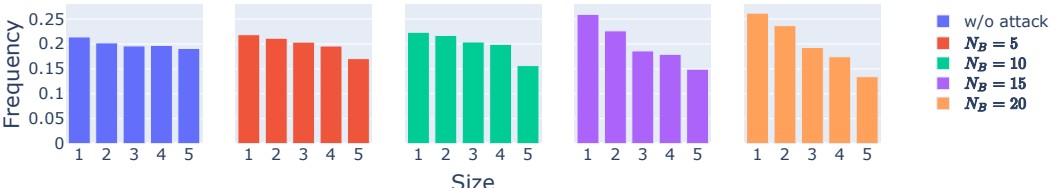

Figure 1: Item distributions in discrete settings with varying values of $N_B$ for different attackers.

tends to select smaller items. Furthermore, as the attack capability increases, the underlying item distribution undergoes more significant changes. Therefore, in our setting, restricting the number of observable items is a practical approach to create permuted new problem instances that are closer to challenging real-world scenarios, as opposed to allowing an unlimited number of observable items.

In Figure 2, the item distributions from time step 1 to time step 20 are illustrated over different problem instances under perturbations from different attackers, using PCT packing policies in the discrete setting with $N_B = 15, 20$. These item distributions not only partially reflect the problem instance distribution, but also reveal the preference of the attacker at each time step. At the beginning, the attacker favors items with smaller sizes, but as the time steps increase, it tends to sample items uniformly. This variation tendency illustrates the behavior of the attacker. However, it does not necessarily imply that rearranging the item sequence in ascending order based on item volumes would be an optimal or even effective attack. As selecting smaller items intuitively reduces the likelihood of forming supportive planes for future incoming items, attackers tend to choose smaller items to minimize the space utilization of the packing policy. Therefore, essentially, the attacker aims to select items that cannot be used to construct a supportive plane, rather than blindly choosing smaller items. This also explains why the probability of larger items in the beginning phase does not decrease to zero in Figure 2. As shown in Figure 3, we use a toy example to demonstrate that rearranging the item sequence in ascending order based on item sizes would not be an optimal or even effective attack. In this example, the container sizes $S^d, d \in \{x, y, z\}$ are set to 2, and the item sequence consists of four items, denoted as $\mathbf{B} = \{b^1, b^2, b^3, b^4\}$. The sizes of the items are as follows: $b^1 = (1 \times 2 \times 2), b^2 = b^3 = (1 \times 1 \times 1)$, and $b^4 = (1 \times 1 \times 2)$. We generate a new item sequence $\mathbf{B}' = \{b^2, b^3, b^4, b^1\}$ by rearranging $\mathbf{B}$ in ascending order based on item sizes. However, this rearrangement is not an effective attack since the optimal solution can still be obtained using a packing strategy. By comparison, moving both $b^1$ and $b^4$ from $\mathbf{B}$ to the middle between $b^2$ and $b^3$ results in $\mathbf{B}''$, which is an effective attack for achieving minimal space utilization. In such a case, the packing agent cannot pack $b^1$ into the container at the third time step.

### A.1.2   Performance of AR2L Algorithm

In the continuous setting, we benchmark four other RL algorithms that are adaptable to online 3D-BPP, including the packing configuration tree (PCT) (Zhao et al., 2022a) method, the CVaR-Proximal-Policy-Optimization (CPPO) (Ying et al., 2022) algorithm, the robust adversarial reinforcement learning (RARL) (Pinto et al., 2017) algorithm, and the robust $f$-divergence MDP (RfMDP) (Ho et al., 2022; Panaganti et al., 2022) algorithm. Since both the exact AR2L (ExactAR2L) and approximate AR2L (ApproxAR2L) algorithms can adjust the value of $\alpha$ to develop policies that vary in their level of robustness, we use different values of $\alpha = 0.3, 0.5, 0.7, 1.0$ to explore the robustness trends with varying values of $\alpha$ in this setting. Likewise, we construct mixture datasets by randomly selecting $\beta\%$ nominal box sequences and rearranging them using the learned corresponding permutation-based attacker for each packing policy.

$Uti.$ Table 2 displays the performance of different robust methods that are compatible with online 3D-BPP on datasets with varying number of challenging problem instances in the continuous setting. ExactAR2L($\alpha$) represents the ExactAR2L algorithm taking different values for $\alpha$, and similarly for ApproxAR2L($\alpha$). In the continuous setting, increasing the value of $\alpha$ can lead to ExactAR2L producing more robust policies against their corresponding permutation-based attackers, as the larger value of $\alpha$ provides more worst-case problem instances during the training of the packing policy. However, they cannot consistently improve their performance on the nominal dynamics. As

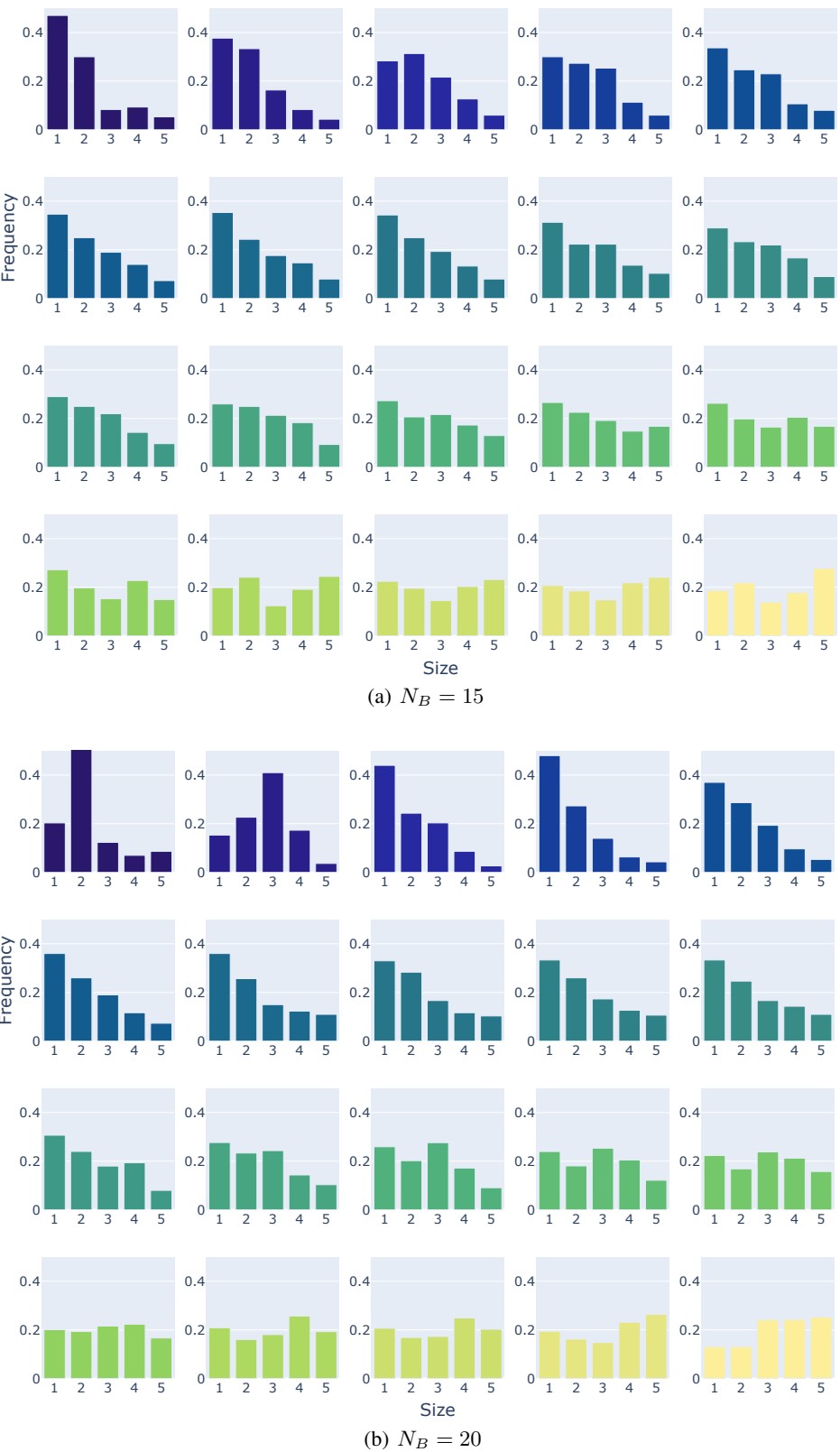

Figure 2: Item distribution at each time step in the discrete setting. From the top left chart to the bottom right chart, it represents the item distribution from time step 1 to time step 20 in sequence.

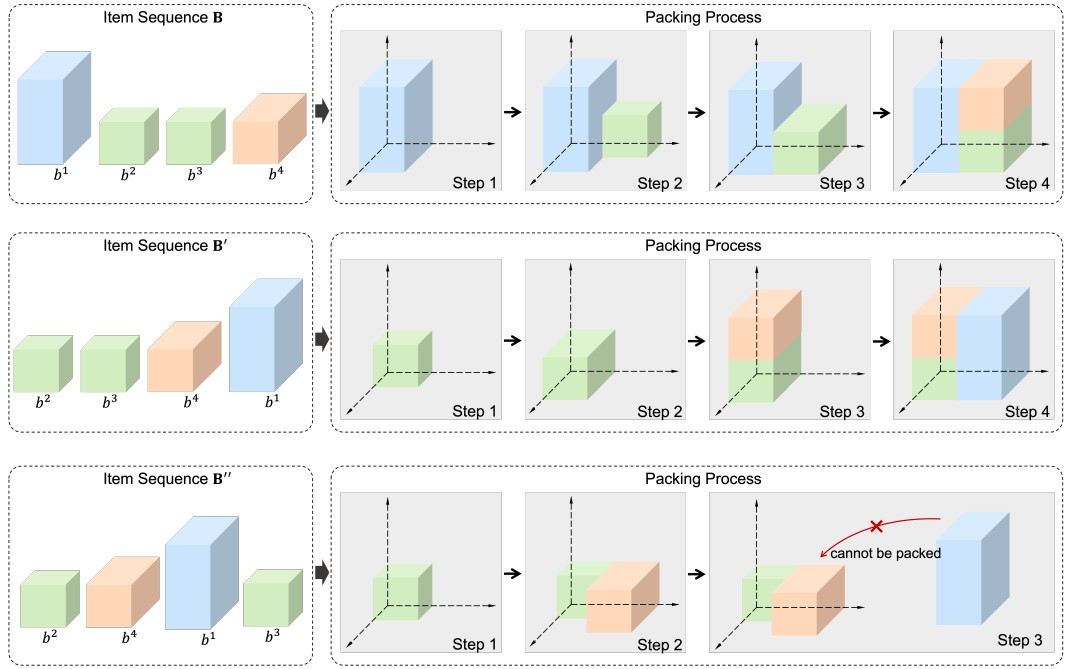

Figure 3: The toy example of packing processes with different item sequences.

the number of observable items of the attacker increases, there is a corresponding increase in the distribution deviation between the perturbed data and the nominal data ($\beta = 0$). For the scenario when $N_B = 5, 10$, the distribution deviation is comparatively small. Thus, it is acceptable to incorporate more challenging instances by increasing the value of $\alpha$ to 0.7. This adjustment will further enhance the generalization of the model and lead to improved performance in terms of space utilization compared to PCT. By comparison, in the cases of $N_B = 15$ and $N_B = 20$, ExactAR2L tends to favor $\alpha = 0.3$ to produce satisfactory performance for $\beta = 0$. Thus, by incorporating the perturbed data with small deviation from the nominal data during training, the generalization of the packing policy can be enhanced, leading to the performance improvement. But the larger deviation can degenerate the performance of the packing policy on the nominal datasets, as demonstrated in RARL. Likewise to the tendency in discrete setting, the ExactAR2L algorithm prefers the larger value of $\alpha$ as the value of $\beta$ increases. Furthermore, due to the value estimation error, ApproxAR2L cannot perform as well as its exact counterpart in most of tasks under the continuous setting. However, to our surprise, we observed that in the case of $N_B = 5$, ApproxAR2L surpasses its corresponding exact counterpart in terms of performance. This could be attributed to the relatively small estimation error resulting from the small deviation in dynamics in this setting, where removing the mixture-dynamics model results in a more stable learning process compared to the exact algorithm. In the remaining tasks, the estimation error in ApproxAR2L can result in more unstable learning, which makes the trend in performance across different values of $\alpha$ less clear compared to the ExactAR2L algorithm.

**Std.** As shown in Table 2, in almost all tasks, ExactAR2L with a larger value of $\alpha$ ($\alpha = 1.0$) exhibits superiority over PCT in terms of $Std.$. Thus, ExactAR2L can indeed improve the robustness in the continuous setting. However, it should be noted that ExactAR2L does not consistently yield smaller values of $Std.$ compared to RARL. This is because the ExactAR2L policy tends to be less conservative than the RARL policy in the majority of tasks. Likewise, ApproxAR2L algorithm prefers larger value of $\alpha$ to consistently achieve smaller $Std.$ in different tasks.

**Num.** In comparison to PCT, the ExactAR2L(1.0) algorithm demonstrates the ability to pack more items in 19 tasks, with a minor decrease observed in 1 task. However, when compared to RARL, the ExactAR2L(1.0) policy only outperforms in 9 tasks. This discrepancy can primarily be attributed to the performance drop observed in the worst-case dynamics of the ExactAR2L algorithm. The ApproxAR2L policy tends to prefer $\alpha = 0.7$ over the RfMDP policy when it comes to packing more items across all the tasks.

Table 2: Performance of robust methods under the perturbation in the continuous setting.

| | Methods | $\beta = 0$ | | | $\beta = 25$ | | | $\beta = 50$ | | | $\beta = 75$ | | | $\beta = 100$ | | |
|---|---|---|---|---|---|---|---|---|---|---|---|---|---|---|---|---|
| | | Uti.(%) | Std. | Num. | Uti.(%) | Std. | Num. | Uti.(%) | Std. | Num. | Uti.(%) | Std. | Num. | Uti.(%) | Std. | Num. |
| $N_B = 5$ | PCT | 65.4 | 9.8 | 24.8 | 60.6 | 12.0 | 23.6 | 56.4 | 11.7 | 22.6 | 52.4 | 11.5 | 21.7 | 48.6 | 7.4 | 21.0 |
| | CPPO | 65.0 | 10.7 | 24.7 | 60.3 | 13.2 | 23.5 | 56.6 | 12.3 | 22.6 | 52.7 | 12.6 | 21.8 | 47.8 | 8.1 | 20.5 |
| | RARL | 65.7 | 10.9 | 25.1 | 62.1 | 12.0 | 24.4 | 57.4 | 11.7 | 23.5 | 53.7 | 11.3 | 22.7 | **50.8** | 8.8 | 22.2 |
| | ExactAR2L(0.3) | 65.9 | 8.8 | 25.3 | 61.6 | 10.7 | 24.2 | 57.7 | 10.5 | 23.3 | 53.5 | 11.5 | 22.4 | 48.9 | 7.7 | 21.3 |
| | ExactAR2L(0.5) | 66.0 | 9.0 | 25.0 | 61.6 | 11.3 | 24.0 | 56.9 | 10.7 | 23.0 | 53.8 | 10.5 | 22.5 | 49.9 | 7.2 | 21.6 |
| | ExactAR2L(0.7) | 67.0 | 9.8 | 25.5 | 62.4 | 12.6 | 24.2 | 57.8 | 11.5 | 23.1 | 54.7 | 10.7 | 22.2 | 50.1 | 7.3 | 21.1 |
| | ExactAR2L(1.0) | 66.7 | 9.6 | 25.4 | 62.6 | 11.3 | 24.6 | 57.9 | 10.3 | 23.6 | **55.1** | 10.0 | 23.1 | 50.3 | 6.0 | 22.1 |
| | RfMDP | 62.9 | 10.4 | 24.0 | 57.2 | 13.3 | 22.7 | 52.4 | 12.5 | 21.6 | 48.7 | 12.4 | 20.9 | 43.3 | 8.6 | 19.7 |
| | ApproxAR2L(0.3) | 66.3 | 9.1 | 25.1 | 62.5 | 11.8 | 24.3 | 57.3 | 11.9 | 23.0 | 52.5 | 11.7 | 22.0 | 48.2 | 7.6 | 21.1 |
| | ApproxAR2L(0.5) | 67.1 | 8.5 | 25.5 | 62.4 | 11.6 | 24.1 | 58.0 | 11.4 | 23.0 | 53.8 | 11.4 | 22.1 | 49.1 | 7.6 | 20.9 |
| | ApproxAR2L(0.7) | **67.4** | 8.8 | 25.6 | **63.3** | 10.9 | 24.6 | **58.4** | 11.0 | 23.5 | 54.1 | 11.6 | 22.5 | 49.5 | 8.1 | 21.2 |
| | ApproxAR2L(1.0) | 64.3 | 9.4 | 24.6 | 60.7 | 11.0 | 23.6 | 57.2 | 11.0 | 22.9 | 53.4 | 11.1 | 22.1 | 49.7 | 8.1 | 21.4 |
| $N_B = 10$ | PCT | 65.3 | 9.6 | 24.8 | 58.9 | 15.0 | 23.4 | 53.1 | 15.4 | 22.2 | 48.0 | 15.4 | 21.2 | 41.4 | 9.4 | 19.7 |
| | CPPO | 65.5 | 10.5 | 24.9 | 59.0 | 15.1 | 23.9 | 52.3 | 15.0 | 22.7 | 46.1 | 14.0 | 21.7 | 39.9 | 6.8 | 20.7 |
| | RARL | 63.4 | 9.1 | 24.1 | 58.6 | 12.1 | 23.6 | 54.7 | 12.2 | 23.3 | 50.5 | 11.3 | 22.9 | **45.9** | 7.0 | 22.5 |
| | ExactAR2L(0.3) | 65.5 | 9.5 | 24.8 | 58.5 | 14.1 | 23.4 | 52.2 | 14.3 | 22.1 | 46.5 | 13.3 | 21.1 | 40.2 | 6.7 | 19.9 |
| | ExactAR2L(0.5) | 66.5 | 8.8 | 25.3 | 59.5 | 14.8 | 23.8 | 54.1 | 14.8 | 22.6 | 47.9 | 14.4 | 21.3 | 41.1 | 8.2 | 19.9 |
| | ExactAR2L(0.7) | **67.8** | 9.0 | 25.8 | **61.8** | 13.9 | 24.7 | **55.2** | 15.6 | 23.2 | 49.4 | 14.8 | 22.0 | 43.0 | 9.6 | 20.7 |
| | ExactAR2L(1.0) | 65.9 | 9.5 | 25.0 | 60.6 | 13.1 | 24.3 | 54.4 | 13.3 | 23.2 | **50.6** | 12.6 | 22.7 | 45.3 | 7.9 | 22.1 |
| | RfMDP | 62.5 | 8.7 | 23.8 | 55.6 | 13.5 | 22.3 | 50.0 | 13.4 | 21.3 | 45.2 | 12.9 | 20.5 | 39.2 | 7.9 | 19.2 |
| | ApproxAR2L(0.3) | 65.5 | 10.0 | 24.9 | 58.3 | 15.0 | 23.5 | 51.4 | 14.1 | 22.0 | 46.1 | 12.7 | 21.1 | 40.7 | 7.5 | 20.2 |
| | ApproxAR2L(0.5) | 65.8 | 9.2 | 24.9 | 59.9 | 14.3 | 23.7 | 54.1 | 15.2 | 22.4 | 48.2 | 14.6 | 21.1 | 40.8 | 8.4 | 19.4 |
| | ApproxAR2L(0.7) | 66.8 | 8.1 | 25.5 | 59.7 | 14.2 | 23.6 | 53.8 | 14.9 | 22.2 | 48.3 | 15.3 | 20.9 | 42.0 | 9.9 | 19.4 |
| | ApproxAR2L(1.0) | 64.7 | 7.2 | 24.6 | 58.5 | 12.5 | 23.7 | 53.2 | 13.0 | 23.1 | 47.3 | 12.6 | 22.3 | 42.6 | 7.9 | 21.9 |
| $N_B = 15$ | PCT | 65.8 | 10.1 | 25.0 | 57.6 | 17.4 | 23.3 | 50.7 | 17.5 | 21.8 | 42.7 | 16.4 | 20.1 | 35.9 | 10.8 | 18.5 |
| | CPPO | 64.6 | 10.2 | 24.6 | 57.4 | 17.1 | 23.1 | 50.0 | 17.9 | 21.2 | 42.2 | 17.7 | 19.5 | 33.5 | 10.4 | 17.6 |
| | RARL | 62.7 | 8.7 | 23.9 | 57.0 | 12.7 | 23.9 | 51.6 | 12.7 | 24.1 | 46.5 | 11.6 | 24.1 | **41.0** | 6.2 | 24.2 |
| | ExactAR2L(0.3) | **66.3** | 8.3 | 25.2 | 59.1 | 15.0 | 23.7 | 51.6 | 16.4 | 22.2 | 45.1 | 15.5 | 20.8 | 37.2 | 8.4 | 19.2 |
| | ExactAR2L(0.5) | 65.2 | 8.9 | 24.8 | 58.3 | 15.6 | 23.6 | 52.0 | 16.2 | 22.5 | 44.8 | 16.3 | 21.0 | 37.7 | 11.1 | 19.7 |
| | ExactAR2L(0.7) | 65.5 | 9.4 | 24.9 | 58.6 | 15.1 | 23.8 | **52.4** | 15.3 | 22.4 | 45.7 | 15.6 | 20.9 | 39.0 | 10.7 | 19.5 |
| | ExactAR2L(1.0) | 65.7 | 10.7 | 25.1 | **59.5** | 15.4 | 24.3 | **52.4** | 16.2 | 23.4 | **47.2** | 15.2 | 22.6 | 40.7 | 10.2 | 21.5 |
| | RfMDP | 62.8 | 10.2 | 23.9 | 55.3 | 17.0 | 22.5 | 47.9 | 17.8 | 20.9 | 41.4 | 17.0 | 19.5 | 34.5 | 11.9 | 17.9 |
| | ApproxAR2L(0.3) | 65.8 | 9.3 | 25.2 | 58.4 | 16.5 | 23.6 | 50.4 | 17.1 | 21.6 | 44.0 | 16.2 | 20.3 | 36.5 | 10.2 | 18.4 |
| | ApproxAR2L(0.5) | 65.8 | 8.8 | 25.0 | 57.4 | 16.6 | 23.6 | 50.0 | 17.2 | 22.4 | 43.7 | 15.7 | 21.5 | 36.8 | 9.2 | 20.2 |
| | ApproxAR2L(0.7) | 65.8 | 8.5 | 24.9 | 58.6 | 15.2 | 23.9 | 51.6 | 15.7 | 22.9 | 45.1 | 14.9 | 22.2 | 37.7 | 8.0 | 21.2 |
| | ApproxAR2L(1.0) | 65.5 | 9.3 | 24.8 | 58.2 | 15.2 | 23.4 | 51.6 | 14.9 | 22.5 | 45.4 | 14.8 | 21.3 | 38.2 | 9.8 | 20.2 |
| $N_B = 20$ | PCT | 65.8 | 10.1 | 25.0 | 57.6 | 17.4 | 23.3 | 50.7 | 17.5 | 21.9 | 42.8 | 16.5 | 20.1 | 35.9 | 10.8 | 18.5 |
| | CPPO | 64.6 | 10.3 | 24.6 | 57.4 | 17.1 | 23.1 | 50.0 | 17.9 | 21.2 | 42.2 | 17.7 | 19.5 | 33.5 | 10.4 | 17.6 |
| | RARL | 62.7 | 8.7 | 23.9 | 57.0 | 12.7 | 23.9 | 51.6 | 12.7 | 24.1 | **46.5** | 11.6 | 24.1 | **41.0** | 6.2 | 24.2 |
| | ExactAR2L(0.3) | **66.9** | 7.7 | 25.3 | 57.2 | 17.3 | 23.2 | 48.7 | 18.7 | 21.6 | 40.1 | 16.5 | 20.0 | 31.9 | 8.5 | 18.5 |
| | ExactAR2L(0.5) | 65.9 | 9.3 | 24.9 | 57.4 | 16.6 | 23.2 | 50.9 | 17.1 | 22.2 | 43.5 | 17.0 | 20.9 | 35.0 | 10.4 | 19.1 |
| | ExactAR2L(0.7) | 65.0 | 9.1 | 24.8 | 57.7 | 16.1 | 23.4 | 49.9 | 17.5 | 21.8 | 42.9 | 16.9 | 20.5 | 35.3 | 11.3 | 18.7 |
| | ExactAR2L(1.0) | 64.9 | 9.2 | 24.5 | 58.1 | 15.6 | 23.6 | **51.3** | 17.2 | 22.4 | 44.6 | 16.9 | 21.3 | 37.4 | 12.0 | 19.9 |
| | RfMDP | 62.8 | 10.2 | 23.9 | 55.3 | 17.0 | 22.5 | 47.9 | 17.8 | 20.9 | 41.4 | 17.1 | 19.5 | 34.5 | 11.9 | 17.9 |
| | ApproxAR2L(0.3) | 64.9 | 9.9 | 24.7 | 55.6 | 18.6 | 22.6 | 47.2 | 19.2 | 20.9 | 39.8 | 18.3 | 19.4 | 30.9 | 9.1 | 17.3 |
| | ApproxAR2L(0.5) | 65.8 | 10.8 | 25.0 | 57.5 | 17.9 | 23.7 | 49.4 | 19.0 | 22.4 | 41.7 | 17.8 | 21.1 | 32.9 | 8.9 | 19.3 |
| | ApproxAR2L(0.7) | 66.7 | 7.7 | 25.3 | **58.4** | 16.7 | 24.2 | 50.3 | 18.0 | 23.0 | 42.3 | 17.2 | 21.9 | 33.6 | 7.4 | 20.3 |
| | ApproxAR2L(1.0) | 64.8 | 9.5 | 24.8 | 58.3 | 15.4 | 24.2 | 50.1 | 17.6 | 23.1 | 42.5 | 16.8 | 21.8 | 34.5 | 8.7 | 20.7 |

**Hyperparameter Configurations.** Based on the observations presented in Table 2, it can be concluded that $\alpha = 1.0$ is the optimal choice for achieving a desired balance between policy performance in both nominal and worst-case scenarios. The ExactAR2L(1.0) algorithm demonstrates competitive robustness and average performance when compared to both PCT and RARL. Compared to the baseline method RfMDP, ApproxAR2L chooses $\alpha = 0.7$ for the best performance.

In addition, although the convergence rate is not our primary concern in evaluating the RL algorithm, we still intend to observe the trend in the learning curves across varying attack capabilities. Our ExactAR2L algorithm shows greater improvement in terms of convergence rate in both the discrete and continuous settings as the number of observable items of the attacker increases, as demonstrated clearly in Figure 4 5.

### A.1.3 Visualization Results

Figure 8 shows a qualitative comparison of the visualized bin packing results obtained by the ExactAR2L, RARL, and PCT policies under the nominal and worst-case dynamics with varying numbers of observable items $N_B = 15, 20$ in the discrete setting. As shown in Figure 8(a) 8(b), the ExactAR2L algorithm allows more smaller margin spaces in the container to minimize sizes of those unoccupied spaces. By comparison, the RARL algorithm attempts to create larger spaces, which may pose a risk of future items not fitting into those spaces. The visualization results of the ExactAR2L and PCT algorithms under their respective worst-case dynamics are depicted in Figure 8(c) and 8(d).

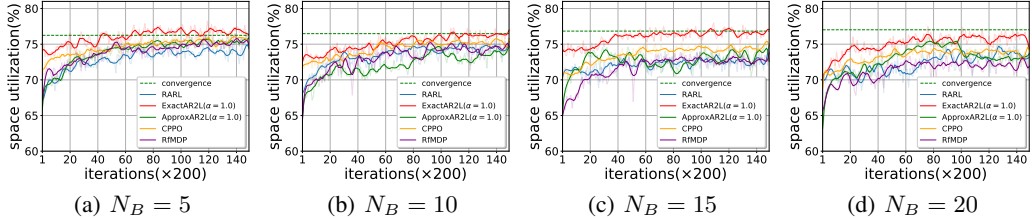

(a) $N_B = 5$      (b) $N_B = 10$      (c) $N_B = 15$      (d) $N_B = 20$

Figure 4: The learning curves of robust RL algorithms for nominal problem instances with different values of $N_B$ in the discrete setting.

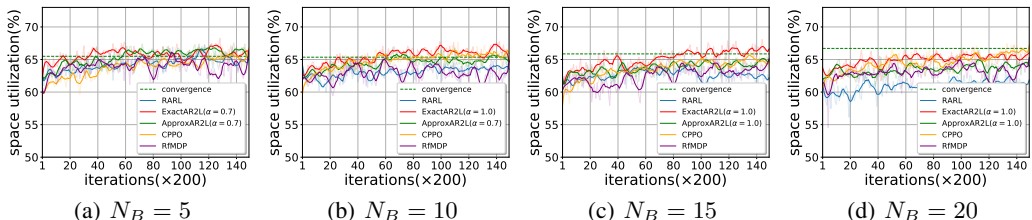

(a) $N_B = 5$      (b) $N_B = 10$      (c) $N_B = 15$      (d) $N_B = 20$

Figure 5: The learning curves of robust RL algorithms for nominal problem instances with different values of $N_B$ in the continuous setting.

One observation is that attackers designed for packing policies tend to prioritize smaller items at the beginning, as a majority of smaller items are likely to occur at the bottom of the container. In such a scenario, the ExactAR2L policy prioritizes the creation of supportive planes for future items, even at the cost of producing smaller margin spaces.

### A.1.4 Baselines Comparisons

Since we implement PCT with different training configurations, we conducted a comparison between our implementation and the official implementation in this section. We conducted the comparison of the algorithms in terms of convergence rate across various values of $N_B$, specifically $N_B = 1, 5, 10, 15, 20$. To ensure a fair and consistent comparison, we employed the same number of problem instances during training. Each iteration utilized a fixed rollout length of 30, and we maintained a constant number of 64 parallel processes throughout the experiments. As shown in Figure 6 and Figure 7, our results clearly demonstrate that our implementation exhibits a faster convergence rate across different settings, considering various numbers of observable items.

### A.1.5 Practicability of AR2L Algorithm

To validate the practicality of exact AR2L algorithm in real-world scenarios, we directly evaluated our model without retraining on the Mixed-item Dataset (MI Dataset) (Yuan et al., 2023; Elhedhli et al., 2019) which follows the generation scheme proposed by Elhedhli et al. (2019) for the realistic 3D-BPP instance generator. MI dataset has 10 thousand items, with 4668 species, and occurrences vary from 1 to 15. The pallet dimensions is set to the size often used in practice: $S^x = 120$, $S^y = 100$, and $S^z = 100$. The results are presented in the Table 3. It is evident that in both settings of $N_B = 15$ and $N_B = 20$, our exact AR2L algorithm demonstrates superior performance compared to PCT and RARL across various metrics in the real-world dataset.

Table 3: The performance of PCT, RARL and exact AR2L evaluated on the Mixed-Item Dataset

| Methods | $N_B = 15$ | | | $N_B = 20$ | | |
|---|---|---|---|---|---|---|
| | $Uti.(\%)$ | $Std.$ | $Num.$ | $Uti.(\%)$ | $Std.$ | $Num.$ |
| PCT | 48.3 | 8.5 | 16.8 | 48.7 | 10.1 | 16.9 |
| RARL | 48.8 | 8.8 | 16.9 | 48.8 | 8.3 | 16.9 |
| AR2L | 50.2 | 8.5 | 17.4 | 52.9 | 6.5 | 18.3 |

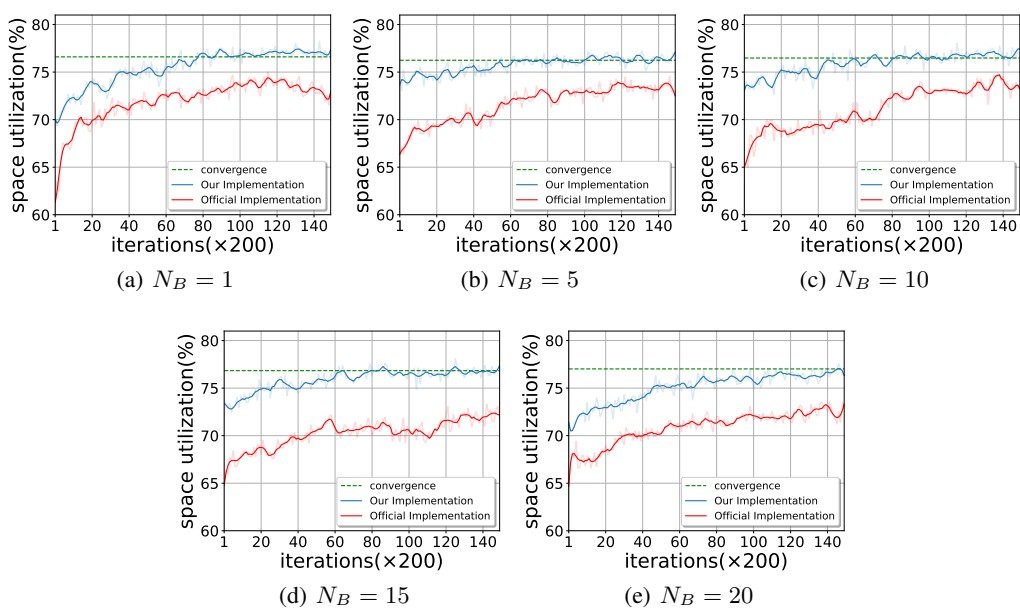

Figure 6: The training curves of our implementation and the official implementation for PCT in the discrete setting.

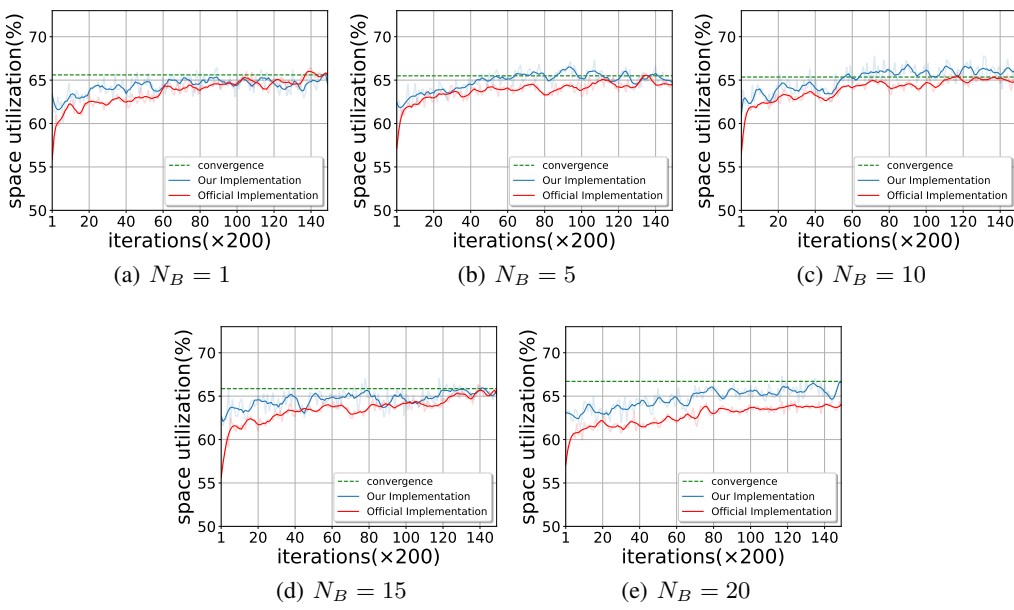

Figure 7: The training curves of our implementation and the official implementation for PCT in the continuous setting.

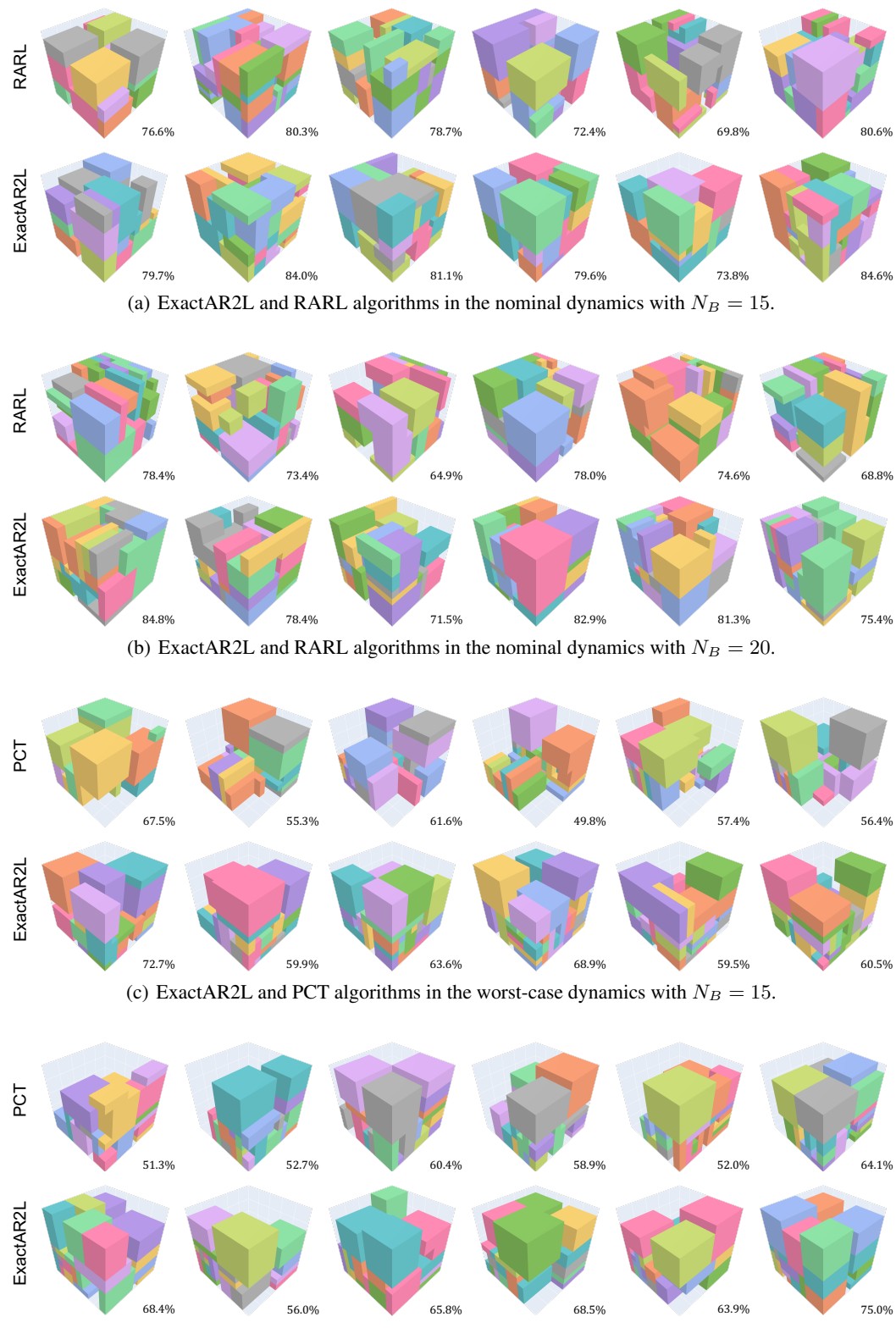

(a) ExactAR2L and RARL algorithms in the nominal dynamics with $N_B = 15$.

(b) ExactAR2L and RARL algorithms in the nominal dynamics with $N_B = 20$.

(c) ExactAR2L and PCT algorithms in the worst-case dynamics with $N_B = 15$.

(d) ExactAR2L and PCT algorithms in the worst-case dynamics with $N_B = 20$.

Figure 8: Visualization results of various methods. The space utilization values for each sub-figure are displayed alongside the respective visualization.

## A.2 Proofs

**Lemma 1.** *[Achiam et al. (2017), Appendix.10.1.1] Given a policy $\pi(\cdot|s)$, a transition model $P(\cdot|s,a)$, and an initial state distribution $\mu(s)$, the discounted state visitation distribution $d_{\pi,P}(s)$ under policy $\pi$ can be described as:*

$$d_{\pi,P}(s) = (1-\gamma)\mu(s) + \gamma\sum_{s',a} d_{\pi,P}(s')\pi(a|s')P(s|s',a) \tag{1}$$

**Theorem 1.** *The model discrepancy between two models can be described as $d(P^1||P^2) \triangleq \max_{s,a} D_{TV}(P^1(\cdot|s,a)||P^2(\cdot|s,a))$. The lower bound for the newly defined objective is derived as follows:*

$$\eta(\pi,P^o) + \alpha\eta(\pi,P^w) \geq (1+\alpha)\eta(\pi,P^m) - \frac{2\gamma|r|_{\max}}{(1-\gamma)^2}(d(P^m||P^o) + \alpha d(P^m||P^w)) \tag{2}$$

*Proof.* Given two dynamics models $P^1(\cdot|s,a)$ and $P^2(\cdot|s,a)$, the absolute value of the difference of returns of a policy $\pi$ can be represented as

$$|\eta(\pi,P^1) - \eta(\pi,P^2)| = |\mathrm{E}_{s\sim\mu(s)}[V^{P^1}(s) - V^{P^2}(s)]| \leq \mathrm{E}_{s\sim\mu(s)}|V^{P^1}(s) - V^{P^2}(s)|, \tag{3}$$

Here, we set $\Delta V(s) = V^{P^1}(s) - V^{P^2}(s)$, and it can be transformed as:

$$\Delta V(s) = \mathrm{E}_{a\sim\pi}[r(s,a) + \gamma\mathrm{E}_{s'\sim P^1}(V^{P^1}(s'))] - \mathrm{E}_{a\sim\pi}[r(s,a) + \gamma\mathrm{E}_{s'\sim P^2}(V^{P^2}(s'))]$$
$$= \gamma\underbrace{\mathrm{E}_{a\sim\pi}\mathrm{E}_{s'\sim P^1}(V^{P^1}(s')) - \gamma\mathrm{E}_{a\sim\pi}\mathrm{E}_{s'\sim P^2}(V^{P^2}(s'))}_{\textcircled{1}} \tag{4}$$

The term $\textcircled{1}$ can be firstly written as:

$$\mathrm{E}_{a\sim\pi}\mathrm{E}_{s'\sim P^1}(V^{P^1}(s')) - \mathrm{E}_{a\sim\pi}\mathrm{E}_{s'\sim P^2}(V^{P^2}(s'))$$
$$=\mathrm{E}_{a\sim\pi}\mathrm{E}_{s'\sim P^1}(V^{P^1}(s')) - \mathrm{E}_{a\sim\pi}\mathrm{E}_{s'\sim P^1}(V^{P^2}(s')) + \mathrm{E}_{a\sim\pi}\mathrm{E}_{s'\sim P^1}(V^{P^2}(s')) - \mathrm{E}_{a\sim\pi}\mathrm{E}_{s'\sim P^2}(V^{P^2}(s'))$$
$$=\mathrm{E}_{a\sim\pi}\mathrm{E}_{s'\sim P^1}\Delta V(s') + \mathrm{E}_{a\sim\pi}\sum_{s'}(P^1(s'|s,a) - P^2(s'|s,a))V^{P^2}(s')$$

$$\tag{5}$$

We then derive an upper bound of the second term in Equation 5 as:

$$\mathrm{E}_{a\sim\pi}\sum_{s'}(P^1(s'|s,a) - P^2(s'|s,a))V^{P^2}(s')$$
$$\leq \max_{s'}V^{P^2}(s')\mathrm{E}_{a\sim\pi}\sum_{s'}(P^1(s'|s,a) - P^2(s'|s,a))$$
$$\leq \frac{2|r|_{\max}}{1-\gamma}\frac{1}{2}\mathrm{E}_{a\sim\pi}\sum_{s'}(P^1(s'|s,a) - P^2(s'|s,a))$$
$$= \frac{2|r|_{\max}}{1-\gamma}\mathrm{E}_{a\sim\pi}D_{TV}(P^1(\cdot|s,a)||P^2(\cdot|s,a)). \tag{6}$$

Next, Equation 5 and 6 can be plugged into Equation 4, written as:

$$\Delta(s) \leq \gamma\mathrm{E}_{a\sim\pi}\mathrm{E}_{s'\sim P^1}\Delta V(s') + \frac{2\gamma|r|_{\max}}{1-\gamma}\mathrm{E}_{a\sim\pi}D_{TV}(P^1(\cdot|s,a)||P^2(\cdot|s,a)) \tag{7}$$

Thus, the expectation of $\Delta(s)$ can be further upper bounded by:

$$\mathrm{E}_{s\sim d_{\pi,P^1}}\Delta(s)$$
$$\leq \mathrm{E}_{s\sim d_{\pi,P^1}}[\gamma\mathrm{E}_{a\sim\pi}\mathrm{E}_{s'\sim P^1}\Delta V(s') + \frac{2\gamma|r|_{\max}}{1-\gamma}\mathrm{E}_{a\sim\pi}D_{TV}(P^1(\cdot|s,a)||P^2(\cdot|s,a))]$$
$$= \gamma\mathrm{E}_{s\sim d_{\pi,P^1}}[\mathrm{E}_{a\sim\pi}\mathrm{E}_{s'\sim P^1}\Delta V(s')] + \frac{2\gamma|r|_{\max}}{1-\gamma}\mathrm{E}_{s\sim d_{\pi,P^1}}[\mathrm{E}_{a\sim\pi}D_{TV}(P^1(\cdot|s,a)||P^2(\cdot|s,a))] \tag{8}$$
$$\leq \gamma\mathrm{E}_{s\sim d_{\pi,P^1}}[\mathrm{E}_{a\sim\pi}\mathrm{E}_{s'\sim P^1}\Delta V(s')] + \frac{2\gamma|r|_{\max}}{1-\gamma}\max_{s,a}D_{TV}(P^1(\cdot|s,a)||P^2(\cdot|s,a))$$

According to Lemma 1, Inequality 8 can be simplified as:

$$\mathrm{E}_{s \sim d_{\pi,P^1}} \Delta(s) \leq \frac{2\gamma |r|_{\max}}{1-\gamma} \max_{s,a} D_{TV}(P^1(\cdot|s,a)||P^2(\cdot|s,a)) + \sum_{s'}[d_{\pi,P^1}(s') - (1-\gamma)\mu(s')]\Delta V(s')$$

(9)

Consequently, $\mathrm{E}_{s \sim d_{\pi,P^1}} \Delta(s)$ can be eliminated, and we can obtain:

$$|\eta(\pi,P^1) - \eta(\pi,P^2)| \leq \mathrm{E}_{s \sim \mu(s)} |V^{P^1}(s) - V^{P^2}(s)| \leq \frac{2\gamma|r|_{\max}}{(1-\gamma)^2} \max_{s,a} D_{TV}(P^1(\cdot|s,a)||P^2(\cdot|s,a))$$

(10)

For the newly defined objective function, we can introduce any unknown dynamics $P^m$ to obtain:

$$\eta(\pi,P^o) - \eta(\pi,P^m) \geq - \max_{s,a} D_{TV}(P^o(\cdot|s,a)||P^m(\cdot|s,a))$$
$$\eta(\pi,P^w) - \eta(\pi,P^m) \geq - \max_{s,a} D_{TV}(P^w(\cdot|s,a)||P^m(\cdot|s,a))$$

(11)

Thus, the lower bound of the objective function can be derived as:

$$\eta(\pi,P^o) + \alpha\eta(\pi,P^w) \geq (1+\alpha)\eta(\pi,P^m) - \frac{2\gamma|r|_{\max}}{(1-\gamma)^2}(d(P^m||P^o) + \alpha d(P^m||P^w)), \quad (12)$$

where $d(P^1||P^2) \triangleq \max_{s,a} D_{TV}(P^1(\cdot|s,a)||P^2(\cdot|s,a))$.

$\square$

**Theorem 2.** *For any given policy $\pi$ and its corresponding worst-case dynamics $P^w$, the adjust Bellman operator $\mathcal{T}_a$ is a contraction whose fixed point is $V_a^\pi$. The operator equation $\mathcal{T}_a V_a^\pi = V_a^\pi$ has a unique solution.*

*Proof.* For any given policy $\pi$ and its corresponding worst-case dynamics $P^w$, the adjustable robust Bellman operator $\mathcal{T}_a$ is defined as:

$$\mathcal{T}_a V_a^\pi(s) = \mathbb{E}_{a \sim \pi}[r(s,a) + \gamma \sup_{P_{s,a}^m \in \mathcal{P}_{s,a}^m} \mathbb{E}_{s' \sim P_{s,a}^m}[V_a^\pi(s')]], \quad (13)$$

where the uncertainty set is defined as $\mathcal{P}^m = \otimes_{(s,a) \in \mathcal{S} \times \mathcal{A}} \mathcal{P}_{s,a}^m$; $\mathcal{P}_{s,a}^m = \{P_{s,a} \in \Delta(\mathcal{S}) : D_{TV}(P_{s,a}||P_{s,a}^o) + \alpha D_{TV}(P_{s,a}||P_{s,a}^w) \leq \rho'\}$.

For any two adjustable robust value functions $V_1$ and $V_2$, we have:

$$||\mathcal{T}_a V_1 - \mathcal{T}_a V_2||_\infty$$
$$= \max_s |\mathbb{E}_{a \sim \pi}[r(s,a) + \gamma \sup_{P_{s,a}^m \in \mathcal{P}_{s,a}^m} \mathbb{E}_{s' \sim P_{s,a}^m}[V_1(s')] - r(s,a) - \gamma \sup_{P_{s,a}^m \in \mathcal{P}_{s,a}^m} \mathbb{E}_{s' \sim P_{s,a}^m}[V_2(s')]]|$$
$$= \gamma \max_s |\mathbb{E}_{a \sim \pi}[\sup_{P_{s,a}^m \in \mathcal{P}_{s,a}^m} \mathbb{E}_{s' \sim P_{s,a}^m}[V_1(s')] - \sup_{P_{s,a}^m \in \mathcal{P}_{s,a}^m} \mathbb{E}_{s' \sim P_{s,a}^m}[V_2(s')]]|$$
$$\leq \gamma \max_s \mathbb{E}_{a \sim \pi}|\sup_{P_{s,a}^m \in \mathcal{P}_{s,a}^m} \mathbb{E}_{s' \sim P_{s,a}^m}[V_1(s')] - \sup_{P_{s,a}^m \in \mathcal{P}_{s,a}^m} \mathbb{E}_{s' \sim P_{s,a}^m}[V_2(s')]|$$
$$= \gamma \max_s \mathbb{E}_{a \sim \pi}|\mathbb{E}_{s' \sim P_{s,a}^{m,1}}[V_1(s')] - \mathbb{E}_{s' \sim P_{s,a}^{m,2}}[V_2(s')]|$$
$$\leq \gamma \max_s \mathbb{E}_{a \sim \pi}|\mathbb{E}_{s' \sim P_{s,a}^{m,1}}[V_1(s') - V_2(s')]|$$
$$\leq \gamma \max_s \mathbb{E}_{a \sim \pi} \mathbb{E}_{s' \sim P_{s,a}^{m,1}}|V_1(s') - V_2(s')|$$
$$\leq \gamma \max_s \mathbb{E}_{a \sim \pi} \max_s |V_1(s') - V_2(s')|$$
$$= \gamma \max_s \mathbb{E}_{a \sim \pi}||V_1 - V_2||_\infty$$
$$= \gamma ||V_1 - V_2||_\infty,$$

(14)

where $P_{s,a}^{m,1} = \arg\sup_{P_{s,a}^m} \mathbb{E}_{s \sim P_{s,a}^m} V_1(s)$ and $P_{s,a}^{m,2} = \arg\sup_{P_{s,a}^m} \mathbb{E}_{s \sim P_{s,a}^m} V_2(s)$.

Let us assume the existence of two different fixed points $V_1^*$ and $V_2^*$ for the adjustable Bellman operator. As both of these are fixed points, the following equality holds:

$$||V_1^* - V_2^*||_\infty = ||\mathcal{T}_a V_1^* - \mathcal{T}_a V_2^*||_\infty \tag{15}$$

According to the proof in 14, the following inequality holds:

$$||\mathcal{T}_a V_1^* - \mathcal{T}_a V_2^*||_\infty \leq \gamma ||V_1^* - V_2^*||_\infty \tag{16}$$

Thus, for both Equation 15 and Equation 16 to strictly hold true simultaneously, it must be the case that $V_1^*$ is equal to $V_2^*$.

$\square$

**Definition 1.** *Let $f : (0, \infty) \to \mathbb{R}$ be a convex function with $f(1) = 1$. Let $P$ and $Q$ be two probability distributions on a measurable space. Then, the $f$-divergence is defined as*

$$D_f(P||Q) = \mathbb{E}_Q[f(\frac{\mathrm{d}Q}{\mathrm{d}P})], \tag{17}$$

*where $\frac{\mathrm{d}P}{\mathrm{d}Q}$ is a Radon-Nikodym derivative.*

**Proposition 1.** *The policy evaluation of AR2L can be formulated as a constrained optimization problem. The objective is to maximize the value function over the uncertainty set $\mathcal{P}^m$, subject to the constraint defined in $\mathcal{P}^m$. By representing the TV distance using the $f$-divergence (Shapiro, 2017), the adjustable robust Bellman operator $\mathcal{T}_a$ can be equivalently written as*

$$\mathcal{T}_a V_a^\pi(s) = \mathbb{E}_{a \sim \pi}[r(s, a) + \frac{\gamma}{1 + \alpha} \inf_{\lambda, \mu_1, \mu_2, \mu} (\mathbb{E}_{s' \sim P_{s,a}^o}[V_a^\pi(s') - \mu_1(s')]_+ +$$
$$\alpha \mathbb{E}_{s' \sim P_{s,a}^w}[V_a^\pi(s') - \mu_2(s')]_+ + \mu + \lambda \rho(1 + \alpha))], \tag{18}$$

*where $[x]_+$ denotes $\max\{x, 0\}$, $\mu_1(s)$ and $\mu_2(s)$ are Lagrangian multipliers for each $s \in \mathcal{S}$; $\mu = \mu_1(s) + \alpha \mu_2(s)$ holds for each $s \in \mathcal{S}$, and $\lambda = \max_s \{V_a^\pi(s) - \mu_1(s), V_a^\pi(s) - \mu_2(s), 0\}$.*

*Proof.* The $f$-divergence can be utilized to express the constraint defined in the uncertainty set $\mathcal{P}^m$ as follows:

$$D_{TV}(P_{s,a}(s')||P_{s,a}^o(s')) + \alpha D_{TV}(P_{s,a}(s')||P_{s,a}^w(s'))$$
$$= \frac{1}{2} \int_{\mathcal{S}} |\mathrm{d}P_{s,a}(s') - \mathrm{d}P_{s,a}^o(s')| + \frac{\alpha}{2} \int_{\mathcal{S}} |\mathrm{d}P_{s,a}(s') - \mathrm{d}P_{s,a}^w(s')|$$
$$= \frac{1}{2} \int_{\mathcal{S}} |\frac{\mathrm{d}P_{s,a}(s')}{\mathrm{d}P_{s,a}^o(s')} - 1| \mathrm{d}P_{s,a}^o + \frac{\alpha}{2} \int_{\mathcal{S}} |\frac{\mathrm{d}P_{s,a}(s')}{\mathrm{d}P_{s,a}^w(s')} - 1| \mathrm{d}P_{s,a}^w \tag{19}$$
$$= \frac{1}{2} \int_{\mathcal{S}} |\zeta^o(s') - 1| \mathrm{d}P_{s,a}^o + \frac{\alpha}{2} \int_{\mathcal{S}} |\zeta^w(s') - 1| \mathrm{d}P_{s,a}^w \leq \rho',$$

where $\zeta : \mathcal{S} \to \mathbb{R}_+$ is the measurable function. $\zeta^o(s) = \frac{\mathrm{d}P(s)}{\mathrm{d}P^o(s)}$ measures the difference between two dynamics at each sample $s \in \mathcal{S}$, similarly for $\zeta^w$.

In the policy evaluation step of the adjustable robust Bellman operator, the goal is to obtain a mixture dynamics $P_{s,a}^m$ that maximizes the adjustable value function while satisfying the constraint defined in Equation 19. We hereby formulate this procedure as a constrained optimization problem, written as:

$$\max_{\zeta^o, \zeta^w} \quad \int_{\mathcal{S}} V(s) \zeta^o(s) \mathrm{d}P^o(s) + \alpha \int_{\mathcal{S}} V(s) \zeta^w(s) \mathrm{d}P^w(s)$$
$$\text{s.t.} \quad \int_{\mathcal{S}} \phi(\zeta^o(s)) \mathrm{d}P^o + \int_{\mathcal{S}} \phi(\zeta^w(s)) \mathrm{d}P^w \leq (1 + \alpha)\rho$$
$$\int_{\mathcal{S}} \zeta^o(s) \mathrm{d}P^o(s) = 1; \quad \alpha \int_{\mathcal{S}} \zeta^w(s) \mathrm{d}P^w(s) = \alpha \tag{20}$$
$$\zeta^o(s) \mathrm{d}P^o(s) - \alpha \frac{1}{\alpha} \zeta^w(s) \mathrm{d}P^w(s) = 0, \forall s \in \mathcal{S},$$

where the value function $V : \mathcal{S} \to \mathbb{R}$ is also a measurable function. We use $P^o$ and $P^w$ to denote $P_{s,a}^o$ and $P_{s,a}^w$ for brevity, $\rho = \frac{\rho'}{1+\alpha}$ is the normalized radius constant, and $\phi(x(s)) = \frac{1}{2}|x(s) - 1|$ is

a convex and lower semicontinuous function. We thus can use the Lagrangian multiplier method to solve the optimization problme in 20. The Lagrangian function is written as:

$$
L(\zeta^o, \zeta^w, \lambda, \mu_1, \mu_2, \mu_3(s)) = \int_{\mathcal{S}} [(V(s) - \mu_1 - \mu_3(s))\zeta^o - \lambda\phi(\zeta^o)]\mathrm{d}P^o
$$
$$
+ \alpha \int_{\mathcal{S}} [(V(s) - \mu_2 + \frac{1}{2}\mu_3(s))\zeta^w - \lambda\phi(\zeta^w)]\mathrm{d}P^w + \lambda\rho(1+\alpha) + \mu_1 + \mu_2\alpha
\tag{21}
$$

The Lagrangian dual of problem 20 is

$$
\inf_{\lambda \geq 0, \mu_1, \mu_2, \mu_3(s)} \sup_{\zeta^o, \zeta^w} L(\zeta^o, \zeta^w, \lambda, \mu_1, \mu_2, \mu_3(s))
\tag{22}
$$

Since the measurable function space $L(\mathcal{S}, \Sigma(\mathcal{S}), P)$ is decomposable, the supremum in 22 can be taken inside the integral (Shapiro, 2017), that is

$$
\sup_{\zeta^o, \zeta^w} L(\zeta^o, \zeta^w, \lambda, \mu_1, \mu_2, \mu_3(s)) = \int_{\mathcal{S}} \sup_{\zeta^o}[(V(s) - \mu_1 - \mu_3(s))\zeta^o - \lambda\phi(\zeta^o)]\mathrm{d}P^o
$$
$$
+ \alpha \int_{\mathcal{S}} \sup_{\zeta^w}[(V(s) - \mu_2 + \frac{1}{2}\mu_3(s))\zeta^w - \lambda\phi(\zeta^w)]\mathrm{d}P^w + \lambda\rho(1+\alpha) + \mu_1 + \mu_2\alpha
\tag{23}
$$

We can observe that the two terms inside the integral are identical to the definition of the conjugate function $\phi^*(y) = \sup_{x \in \mathbb{R}_+}\{xy - \phi(x)\}$. And $\phi(x)$, as a convex, semicontinuous function, is defined as $\frac{1}{2}|x(s) - 1|$, resulting in $\phi^*(y) = \max\{y, -\frac{1}{2}\}$. The dual problem can be transformed as:

$$
\inf_{\lambda \geq 0, \mu_1, \mu_2, \mu_3(s)} L(\lambda, \mu_1, \mu_2, \mu_3(s)) = \inf_{\lambda \geq 0, \mu_1, \mu_2, \mu_3(s)} \int_{\mathcal{S}} \max\{V(s) - \mu_1 - \mu_3(s), -\frac{1}{2}\lambda\}\mathrm{d}P^o
$$
$$
+ \alpha \int_{\mathcal{S}} \max\{V(s) - \mu_2 + \frac{1}{\alpha}\mu_3(s), -\frac{1}{2}\lambda\}\mathrm{d}P^w + \lambda\rho(1+\alpha) + \mu_1 + \mu_2\alpha
\tag{24}
$$

We use $[x]_+$ to represent $\max\{x, 0\}$ for brevity. It is then transformed as

$$
\inf_{\lambda \geq 0, \mu_1, \mu_2, \mu_3(s)} L(\lambda, \mu_1, \mu_2, \mu_3(s)) = \inf_{\lambda \geq 0, \mu_1, \mu_2, \mu_3(s)} \int_{\mathcal{S}} [V(s) - \mu_1 - \mu_3(s) + \frac{1}{2}\lambda]_+\mathrm{d}P^o
$$
$$
+ \alpha \int_{\mathcal{S}} [V(s) - \mu_2 + \frac{1}{\alpha}\mu_3(s) + \frac{1}{2}\lambda]_+\mathrm{d}P^w + \lambda\rho(1+\alpha) + \mu_1 - \frac{\lambda}{2} + (\mu_2 - \frac{\lambda}{2})\alpha
\tag{25}
$$

Next, we set $\mu_1'(s) = \mu_1 + \mu_3(s) - \frac{1}{2}\lambda$ and $\mu_2'(s) = \mu_2 - \frac{1}{\alpha}\mu_3(s) - \frac{1}{2}\lambda$. And we observe that $\forall s \in \mathcal{S}, \mu_1'(s) + \alpha\mu_2'(s) = \mu_1 - \frac{\lambda}{2} + (\mu_2 - \frac{\lambda}{2})\alpha$. We thus set $\mu = \mu_1'(s) + \alpha\mu_2'(s)$.

$$
\inf_{\lambda \geq 0, \mu_1, \mu_2, \mu_3(s)} L(\lambda, \mu_1, \mu_2, \mu_3(s))
$$
$$
= \inf_{\lambda \geq 0, \mu_1, \mu_2, \mu_3(s)} \int_{\mathcal{S}} [V(s) - \mu_1'(s)]_+\mathrm{d}P^o + \alpha \int_{\mathcal{S}} [V(s) - \mu_2'(s)]_+\mathrm{d}P^w + \lambda\rho(1+\alpha) + \mu
\tag{26}
$$

In addition, based on the analysis of the conjugate function $\phi^*(y)$ in (Panaganti et al., 2022), we can derive the following two inequalities:

$$
\frac{V(s) - \mu_1 - \mu_3(s)}{\lambda} \leq \frac{1}{2}; \quad \frac{V(s) - \mu_2 + \frac{1}{2}\mu_3(s)}{\lambda} \leq \frac{1}{2}
\tag{27}
$$

We thus can obtain $\lambda = \max_s\{V_a^\pi(s) - \mu_1(s), V_a^\pi(s) - \mu_2(s), 0\}$. Building on the proofs above, we can define the adjustable robust Bellman operator as shown in Equation 18.

$\square$

## A.3 Pseudocode of Algorithms

The pseudocodes for both the exact AR2L algorithm and the approximate AR2L algorithm are presented in Algorithm 1 and Algorithm 2, respectively. Note that in practice, the Kullback-Leibler (KL) divergence is adopted to constrain the deviation between distributions in the exact AR2L algorithm.

**Algorithm 1** Exact AR2L Algorithm

---

**Initialize**: packing policy $\pi_{\text{pack}}$, permutation-based attacker $\pi_{\text{perm}}$, mixture-dynamics model $\pi_{\text{mix}}$, and their corresponding value functions $V^{\pi_{\text{pack}}}$, $V^{\pi_{\text{perm}}}$, $V^{\pi_{\text{mix}}}$;
**Input**: stationary item distribution $p_b$;
**Parameter**: robustness weight $\alpha$, number of observable items $N_B$;
**Output**: packing policy $\pi_{\text{pack}}$;

1: **for** $i = 0$ to $max\_iteration$ **do**
2:     # train the permutation-based attacker
3:     Reset the environment and observe $\mathbf{C}_0, \mathbf{B}_0 \sim p_b$;
4:     **for** $t = 0$ to $max\_step$ **do**
5:        Get permuted item sequence: $\mathbf{B}'_t = \pi_{\text{perm}}(\mathbf{C}_t, \mathbf{B}_t)$;
6:        Get location for $b'_{t,1}$: $l_t = \pi_{\text{pack}}(\mathbf{C}_t, \mathbf{B}'_t)$;
7:        Pack $b'_{t,1}$ into the bin, and observe $\mathbf{C}_{t+1}, \mathbf{B}_{t+1} \sim p_b$, reward $r_t^{\text{pack}}$, termination $d_t$;
8:        **if** $d_t == $ True **then**
9:          Update $\pi_{\text{perm}}$ on episode samples to maximize $-\sum r_t^{\text{pack}}$;
10:        **end if**
11:     **end for**
12:     # train the mixture-dynamics model and the packing policy
13:     Reset the environment and observe $\mathbf{C}_0, \mathbf{B}_0 \sim p_b$;
14:     **for** $t = 0$ to $max\_step$ **do**
15:        Get permuted item sequence: $\mathbf{B}''_t = \pi_{\text{mix}}(\mathbf{C}_t, \mathbf{B}_t)$;
16:        Get location for $b''_{t,1}$: $l_t = \pi_{\text{pack}}(\mathbf{C}_t, \mathbf{B}''_t)$;
17:        Pack $b''_{t,1}$ into the bin, and observe $\mathbf{C}_{t+1}, \mathbf{B}_{t+1} \sim p_b$, reward $r_t^{\text{pack}}$, termination $d_t$;
18:        **if** $d_t == $ True **then**
19:          Update $\pi_{\text{mix}}, V^{\pi_{\text{mix}}}$ on episode samples to maximize
            $\sum r_t^{\text{pack}} - (D_{KL}(\pi_{\text{mix}} || \mathbb{1}_{\{x=b_{t+1,1}\}}) + \alpha D_{KL}(\pi_{\text{mix}} || \pi_{\text{perm}}))$;
20:          Update $\pi_{\text{pack}}, V^{\pi_{\text{pack}}}$ on episode samples to maximize $\sum r_t^{\text{pack}}$;
21:        **end if**
22:     **end for**
23: **end for**

---

### A.4 Implementation Details

To implement a packing policy that can work in both the discretized grid world and the continuous solution space, we use the packing configuration tree (PCT)(Zhao et al., 2022a) to represent the state, and employ a transformer(Vaswani et al., 2017) network as the packing policy. PCT is utilized to represent the bin configuration and available empty spaces for the most recently packed item, while the versatile transformer network is capable of handling the variable number of nodes in PCT. We adopt the same network configuration as reported in PCT for the packing policy, with the addition of positional encoding to encode the position information of each item in the observed item sequence.

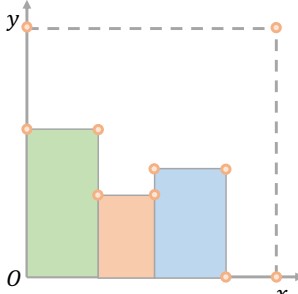

Figure 9: In the $xoy$ plane, circles represent the potential positions generated by the IPs heuristics.

In addition, those available spaces are generated using a heuristic method that takes into account the packing constraints and preferences for the item to be packed. Specifically, in the discrete setting, we utilize the empty maximal spaces (EMSs) (Ha et al., 2017) heuristics to generate potential positions, and subsequently apply the same constraints used in PCT to check the stability of these positions, thereby identifying the feasible positions $\mathbf{L}$. In the continuous setting, we propose a new heuristic methods called Intersection Points (IPs) to generate potential positions. Given the 2D packed items in the $xoy$ plane, the intersection points are generated from the intersection of the sides of cuboid-shaped items and the bin, as shown in Figure 9. To extend this method from 2D to 3D cases, we allow candidate positions to be obtained from the intersection points of the sides in the top view.

---
**Algorithm 2** Approximate AR2L Algorithm
---
**Initialize**: packing policy $\pi_{\text{pack}}$, permutation-based attacker $\pi_{\text{perm}}$, and their corresponding value functions $V_a^{\pi_{\text{pack}}}$, $V^{\pi_{\text{perm}}}$;
**Input**: stationary item distribution $p_b$;
**Parameter**: robustness weight $\alpha$, number of observable items $N_B$;
**Output**: packing policy $\pi_{\text{pack}}$;

 1: **for** $i = 0$ to $max\_iteration$ **do**
 2:     # train the permutation-based attacker
 3:     Reset the environment and observe $\mathbf{C}_0, \mathbf{B}_0 \sim p_b$;
 4:     **for** $t = 0$ to $max\_step$ **do**
 5:         Get permuted item sequence: $\mathbf{B}'_t = \pi_{\text{perm}}(\mathbf{C}_t, \mathbf{B}_t)$;
 6:         Get location for $b'_{t,1}$: $l_t = \pi_{\text{pack}}(\mathbf{C}_t, \mathbf{B}'_t)$;
 7:         Pack $b'_{t,1}$ into the bin, and observe $\mathbf{C}_{t+1}, \mathbf{B}_{t+1} \sim p_b$, reward $r_t^{\text{pack}}$, termination $d_t$;
 8:         **if** $d_t ==$ True **then**
 9:             Update $\pi_{\text{perm}}$ on episode samples to maximize $-\sum r_t^{\text{pack}}$;
10:         **end if**
11:     **end for**
12:     # train the packing policy
13:     Reset the environment and observe $\mathbf{C}_0, \mathbf{B}_0 \sim p_b$;
14:     **for** $t = 0$ to $max\_step$ **do**
15:         Get location for $b_{t,1}$: $l_t = \pi_{\text{pack}}(\mathbf{C}_t, \mathbf{B}_t)$;
16:         Pack $b_{t,1}$ into the bin, and observe $\mathbf{C}_{t+1}, \mathbf{B}_{t+1} \sim p_b$, reward $r_t^{\text{pack}}$, termination $d_t$;
17:         **if** $d_t ==$ True **then**
18:             Permute $\mathbf{B}_t, t \in [0, T]$ to $\mathbf{B}'_t, t \in [0, T]$ using $\pi_{\text{perm}}$;
19:             Use 18 to compute target values for adjustable Bellman values on
                $\{\mathbf{C}_t, \mathbf{B}_t, \mathbf{B}'_t, l_t, r_t^{\text{pack}}, d_t\}, t \in [0, T]$;
20:             Update $\pi_{\text{pack}}, V^{\pi_{\text{pack}}}$ on episode samples to maximize $\sum r_t^{\text{pack}}$;
21:         **end if**
22:     **end for**
23: **end for**
---

In our implementations of both the permutation-based attacker and the mixture-dynamics model, we utilize the transformer (Vaswani et al., 2017) network to model these policies. This choice is motivated by the variable number of observed boxes in different real packing systems, as well as the dynamic changes in the number of packed items during each episode. The transformer is capable of capturing long-range dependencies in spatial data and sequential data. Specifically, we first use two independent element-wise fully-connected (FC) layers to project heterogenous elements of the state into homogenous features. Then, the scaled dot-product attention is relied to calculate the weighted sum of features based on the relevance between given input features. Next, we use the skip-connection operation and the element-wise feed-forward FC layer to obtain the final features $\boldsymbol{x} = \{\boldsymbol{x}_C, \boldsymbol{x}_B\} \in \mathbb{R}^{d_x \times N}$, where $\boldsymbol{x}_C \in \mathbb{R}^{d_x \times N_C}$ and $\boldsymbol{x}_B \in \mathbb{R}^{d_x \times N_B}$ denote features of $\mathbf{C}_t$ and $\mathbf{B}_t$ respectively, $d_x$ is the feature size, and $N = N_C + N_B$ denotes the variable feature number. Finally, the probability distribution over $\mathbf{B}_t$ is described as

$$\pi_{\text{perm}}(\mathbf{B}_t | \mathbf{C}_t, \mathbf{B}_t) = \text{softmax}(c_{\text{temp}} \cdot \tanh(y)), \quad y = \frac{\bar{x}^T \boldsymbol{x}_B}{\sqrt{d_x}} \in \mathbb{R}^{N_B}, \tag{28}$$

where $\bar{x} = \frac{1}{N} \sum_{i=1}^{N} x_i$ and $c_{\text{temp}}$ is the temperature constant.

As the number of packed items $\mathbf{C}$ and the number of available feasible positions $\mathbf{L}$ both vary at different time steps and have irregular shapes, we combine this data into one batch by filling $\mathbf{C}$ and $\mathbf{L}$ to fixed lengths of 80 and 120 in the discrete setting, and 100 and 120 in the continuous setting, respectively, using dummy elements. Each valid element in $\mathbf{C}$ is a vector that specifies the size and position of an item that has already been packed. Each vector in $\mathbf{B}$ represents the sizes of an item that has yet to be packed. Each vector in $\mathbf{L}$ contains the position of a feasible empty space that can accommodate a new item. To permute the observed item sequence, we feed both $\mathbf{C}$ and $\mathbf{B}$ to a transformer network that serves as the attacker. In this transformer network, the sizes of the two element-wise fully connected (FC) layers are fixed at 64. Additionally, there is one attention

layer with one head attention, and the sizes of the query, key, and value vectors are set to 64. The size of the element-wise feed-forward FC layer is also set to 64. In our experiment, we utilize the PPO (Schulman et al., 2017) algorithm to effectively train the packing policy, permutation-based attacker, and mixture-dynamics model. PPO employs multiple parallel processes (64 in this case) to interact with their respective environments and collect data. The rollout length for each iteration is set to 30, and the learning rate is set to 0.0003. To ensure a fair comparison, we keep the same hyperparameter settings mentioned above for each of the methods that we have reproduced.

## A.5 Related Work

**Heuristic Methods.** In heuristic methods proposed for solving 3D-BPP, the score function is often designed to rank item placements based on expert knowledge, which can represent the packing requirements and preferences to some extent. Ha et al. (2017) relied on the deep-bottom-left (DBL) (Karabulut and İnceoğlu, 2005) rule to sort empty maximal spaces (EMSs) (Parreño et al., 2008) for the current item. The best-match-first strategy (Li and Zhang, 2015) gives priority to feasible placements that have the smallest margin for the observed item. Hu et al. (2017) introduced the least surface area heuristic, which selects the maximal empty space and orientation that result in the smallest surface area. The Heightmap-Minimization method (Wang and Hauser, 2019) favors items placement that result in the smallest occupied volume as observed from the loading direction. The maximize-accessible-convex-space method, as described by Hu et al. (2020), aims to optimize the empty space available for future, potentially large items. Despite their simplicity and effectiveness, these methods find difficulties in handling complex packing preferences and adapting to diverse scenarios for solving online 3D-BPP. This is because their packing strategies rely heavily on expert experience and may not be flexible enough to accommodate different situations.

**DRL-based Methods.** To further develop highly effective policies, DRL-based methods are broadly proposed by formulating the online 3D-BPP as a sequential decision-making problem. In this formulation, the state usually includes the bin configuration, which contains information about both the container and the packed items, as well as the observed item on the conveyor. To learn a DRL-based policy, the container is discretized to represent the bin configuration, and a convolutional neural network (CNN) is served as a policy (Hu et al., 2017; Verma et al., 2020; Zhao et al., 2021, 2022b; Yang et al., 2021; Song et al., 2023). However, these learning-based methods only work in a grid world with limited spatial discretization accuracy, which reduces their practical applicability. To deploy DRL-based methods on solving online 3D-BPP with continuous solution space, Zhao et al. (2022a) utilized the Packing Configuration Tree (PCT) to represent the bin configuration and potential available spaces. They then employed a graph attention network (Veličković et al., 2017) to encode the PCT and observed boxes, to encode the PCT and observed boxes, and to select a location from available spaces generated for the most preceding item. However, these methods often focus on optimizing average performance and do not account for worst-case scenarios.

