# OpenReview forum: "Adjustable Robust Reinforcement Learning for Online 3D Bin Packing"
_NeurIPS.cc/2023/Conference — NeurIPS 2023 poster_

### Official Review · Reviewer_Ua6d · 2023-07-05

**Soundness:** 2 fair
**Presentation:** 3 good
**Contribution:** 3 good
**Rating:** 4
**Confidence:** 4

**Summary:**

This work proposed a novel adjustable and robust Reinforcement Learning framework for Online 3D Bin Packing task. The proposed method can achieve balance between performance and the worst-case environment.

**Strengths:**

The writing is clear, and the experimental results demonstrate that the proposed method works well compared with other RL methods for the targeted 3D Bin Packing problem.

**Weaknesses:**

- In the section of “Training and Evaluation Setting”, the author mentioned that they are using two different settings which are discrete and continuous. However, I did not find the table showing the experimental results using continuous setting.

- The author mentioned that they use three metrics(U_ti, Std and Num) to evaluate the performance. But only the best U_ti are highlighted in Table 2. I do think the analysis for Table 2 needs more details especially for the Std and Num.

- The computation complexity is also crucial for the RL algorithms. The comparison of computation complexity should be analyzed.

**Questions:**

- Table 2 is the main result, where larger N_B and larger beta represent the harder problem, and we can see that throughout the table, the baseline RARL often achieves the best results, and the proposed Exact AR2L and Approx AR2L is inferior. What is your interpretation?

- How to ensure the Approx AR2L can approximate the AR2L in the "real-world" setting? Since you do not perform experiments in the real world? I mean, can your added randomness correspond to a real-world robot arm use case (in some industry), and if so, what kind of distribution can characterize such randomness? I believe this must be addressed by data-driven experiments; however I do not see this in your paper.

- Attacker: I understand the "attacker" is used to generate proper samples to train your DRL algorithm for policy improvement. However, do you have a better name for such adversarial or adversary other than an attacker, as they are used in a normal setting. If there is a true attacker of your DRL training, they will probably attack in other places the let the framework down. I understand this might be due to the term already used in the literature, I just want to not confusing the reader about technical terms.

**Limitations:**

There is no such limitation in this aspect for this paper.

---

> ### Author Rebuttal · Authors · 2023-08-09
>
> We highly appreciate the valuable feedback provided by the reviewer. We have carefully considered these concerns, and we would like to address them in the following responses.
>
> Q1: Continuous setting results
>
> Due to space limit in the main text, for the experimental results under the continuous setting where item size follows a continuous distribution, please refer to the Appendix 1.1.2 in our supplementary material.
>
>
> Q2: Experiment results clarification
>
> We thank the reviewer for looking into details of our algorithm performance. As illustrated in Table 2, the ExactAR2L algorithm demonstrates its superiority over PCT with smaller standard deviation (Std) in 17 tasks, while producing a slightly larger Std in only 3 tasks, which demonstrates that ExactAR2L can indeed improve the robustness of the packing policy. Since ExactAR2L is trained on instances from both the nominal and worst-case environments, while RARL is trained only on the worst-case dynamics, the resulting ExactAR2L policy is less conservative than the RARL policy. While the conservativeness of RARL may result in smaller Std in most tasks, it produces worse results when given nominal instances compared to AR2L. It is worth noting that compared to other methods, the value of Std from ExactAR2L is the closest to that of RARL. This observation tells us our ExactAR2L framework can trade off between conservative and risky behavior, as the Std from ExactAR2L is between that of RARL and PCT. Compared to the baseline method RfMDP, ApproxAR2L with $\alpha=0.5$ has a smaller Std in half of the tasks. This is because, similar to ExactAR2L, ApproxAR2L is less conservative than RfMDP, which results in the fact that ApproxAR2L cannot achieve a smaller Std in all tasks.
>
>
> As illustrated in Table 2, the ExactAR2L algorithm can pack more items in 17 tasks compared to PCT, and shows a slight drop in only 3 tasks, where the average drop is 0.2. We found that to achieve a higher score in terms of Num, ExactAR2L consistently favors $\alpha=1.0$ across various tasks of different $N_B$ and $\beta$. Compared to RARL, the ExactAR2L algorithm with $\alpha=1$ can pack at least the same number of items in 16 tasks, and shows a slight drop in 4 tasks. We thus conclude that ExactAR2L with $\alpha=1$ can produce competitive results compared to both RARL and PCT in terms of Num. Compared to the baseline method RfMDP, ApproxAR2L with $\alpha=0.5$ can pack more items in 16 tasks, and shows a slight drop in 4 tasks, where the average drop is 0.25. In the revised paper, we would take the reviewer's suggestions and give more detailed and comprehensive discussions for the simulation results.
>
>
>
> Q3: Computation complexity of AR2L
>
> In our conclusion and limitation section, we highlighted that the ExactAR2L algorithm introduces additional computational complexity in the training phase due to the mixture-dynamics attacker. However, in the bin packing research community and industrial applications, researchers are more concerned about the computational complexity during the inference phase, as we require the packing strategy to efficiently determine the location for each item. From this perspective, AR2L does not introduce additional complexity during inference compared to RARL and PCT.
>
> Q4: AR2L's performance compared to RARL
>
> The AR2L policy is trained on instances from both the worst-case and the nominal dynamics. When evaluating the scenario of $\beta=100$, due to the deviation between the data distributions used for training and testing, AR2L may not consistently outperform RARL. However, the objective of our paper is to *strike a balance between the policy's performance in average and worst-case environments*. As such, for $\beta=0, 25, 50, 75$, AR2L outperforms RARL, demonstrating its superiority in achieving such balancing goal.
>
>
> Q5: Real world setting
>
> We would like to clarify that our work focuses specifically on the bin packing problem. Our goal is to develop a packing policy that assigns a position for each item in a container to ensure the high space utilization. Our focus is not on the manipulation or locomotion of robots, which is a separate module from the location assignment problem of bin packing. The location assignment problem is commonly viewed as a combinatorial optimization (CO) problem, while robotics planning and control typically address the manipulation and locomotion of robots. In the bin packing research community, the real world data is well simulated by the item sequence environment, rather than using mujoco-like robotic environments for dynamics modeling. And we follow the exact setup as bin packing research community to *setup the environment to cover the real-world cases. Furthermore, our study is specifically designed to improve the robustness of the RL-based policy in a CO problem against the common randomness in the permutation of item sequence, rather than perturbations added to the robot arm*. We hope that this clarifies the scope of our research and the specific problem we are addressing. In addition, to validate the practicality of AR2L in real-world scenarios, we directly evaluated it on the Mixed-item Dataset (MI Dataset) [1] which follows the generation scheme for the realistic 3D-BPP instance. We apologize that due to limited space we have included the results on the MI dataset to Table 1 of the submitted PDF file of the global response. Upon analyzing the results, we observe that our AR2L approach outperforms both PCT and RARL across various metrics in the real-world MI dataset
>
>
> Q6: Naming of adversarial policy
>
> Thank you for your suggestions regarding the name of the "attacker". We are considering using a more appropriate name to avoid any confusion.
>
> [1] Samir Elhedhli, Fatma Gzara, and Burak Yildiz. Three-dimensional bin packing and mixed-case palletization. INFORMS Journal on Optimization 2019.

---

> > ### Comment · Reviewer_Ua6d · 2023-08-17
> > **Comments after rebuttal**
> >
> > I acknowledges that I have read the author's rebuttal. I believe it has answered my questions. I think overall this is a paper with sufficient amount of efforts.

---

> > > ### Author Response · Authors · 2023-08-18
> > >
> > > We thank the reviewer again for taking a careful review of our work, and we will incorporate the suggestions into the revised paper. We would appreciate that if the reviewer could re-evaluate the review score.

---

### Official Review · Reviewer_HsT1 · 2023-07-06

**Soundness:** 3 good
**Presentation:** 2 fair
**Contribution:** 3 good
**Rating:** 5
**Confidence:** 4

**Summary:**

For solving the online 3D Bin Packing Problem (BPP), the authors employ an iterative procedure to search for relevant hybrid dynamics and refine their corresponding strategies. By optimizing the weighted sum of returns, AR2L algorithm which improves the robustness of the packing policy achieves a balance between nominal insurance performance and worst case insurance performance.

**Strengths:**

In experiment, the authors evaluate the robustness of six heuristic method. Moreover, this paper gives the exact AR2L algorithm and Approximate AR2L algorithm which has positive results. This two part of solving online 3D Bin Packing Problem has been proved to have relatively good results theoretically and experimentally.

**Weaknesses:**

Experimentation is not sufficient. Continuously valued perturbations for addition may not correspond to distributions in the physical world. Is there any difference between the response of the baseline to the disturbance of these continuous values and the method in the paper to the disturbance of these continuous values. Whether there is a simulation or a robotic arm, a demo for dealing with extreme situations.

**Questions:**

1.	For Figure 2, it should be ensured that the positions of the same modules remain unchanged, and the comparison changes should be highlighted, so as to achieve of the exact and approximate AR2L algorithm
2.	The relationship between offline and online BPP introduced by introduce should be clarified, if not needed, offline should be deleted. Because the whole article does not focus on online BPP
3.	At the same time, please supplement the experimental part in weaknesses


**Limitations:**

due to the mixture-dynamics attacker, it will introduce additional calculations to increase complexity. At the same time for the latency of the whole process will be increased as well.

---

> ### Author Rebuttal · Authors · 2023-08-09
>
> We would like to express our gratitude to the reviewer for providing valuable feedback. We appreciate the reviewer's insights, and we would like to address the concerns raised.
>
>
> Q1: Illustration of schematic figure
>
> Thank you for your valuable suggestions. We agree that it can make the comparison in Figure 2 more clear, and we appreciate your feedback. We will refine the two schematic diagrams in Figure 2 to better align with your suggestions.
>
>
> Q2: Offline and online BPP
>
> Thank you for your feedback regarding the introduction of the relationship between offline BPP and online BPP in our paper. We appreciate your input and understand your point that our focus is on the online BPP rather than the offline BPP.
>
> We included a brief introduction to the background of offline BPP as it is the foundation of the online BPP. However, we understand that it may not be directly relevant to the main focus of our paper. Therefore, we will refine this section as per your suggestion to streamline the paper and better align with our focus on the online BPP.
>
>
> Q3: Experiments on continuously valued perturbations
>
> Thank you for your comments on the experiment section. Since the main focus of our paper is the *perturbations from the permutation of item sequence in the online 3D-BPP*, and we also considered that adding continuous-value noise may not correspond to real-world perturbations, given that the state is defined by the bin configuration and the currently observed item sequences. Therefore, we did not conduct experiments on continuously valued perturbations in our paper. However, we recognize the value of such experiments in extending our framework and better understanding the robustness of learning-based policies. As such, we supplement our paper with an additional experiments on continuously valued perturbations, as per your suggestion. Please refer to the global response for detailed simulation results.
>
> In this experiment, we examine the performance of three different approaches, namely PPO, RARL, and our AR2L method. We conduct this study in the CartPole environment, where we introduce the continuously valued perturbation to the "gravity" parameter to investigate the robustness. The robustness to changes in the 'gravity' parameter is a well-established aspect of robust RL [1]. During the training phase, RARL trains a worst-case attacker to select a continuously valued perturbation to add to the 'gravity' parameter in the environment. On the other hand, AR2L trains a mixture-dynamics attacker to balance the worst-case dynamics derived from the worst-case attacker and the nominal dynamics. In the evaluation phase, we introduce various perturbations to the environment to test the performance of the different approaches across different settings. Please refer to Figure 1 from our submitted PDF file in the global response for the detailed experiment results. Upon analyzing the results, we observe that when larger perturbations (larger absolute x-axis values) are introduced, both AR2L and RARL consistently outperform PCT. However, as the perturbation decreases, RARL's performance shows a significant decline. On the other hand, AR2L's performance lies between that of PCT and RARL, indicating a more balanced and robust performance across varying perturbation levels.
>
> To further validate the practicality of the AR2L algorithm in real-world scenarios, we directly evaluated our model on the Mixed-item Dataset (MI Dataset) which follows the generation scheme proposed by Elhedhli et al. [2] for the realistic 3D-BPP instance generator. MI dataset has 10 thousand items, with 4668 species, and occurrences vary from 1 to 15. The bin dimensions is set to the size often used in practice: L = 120, W = 100, and H = 100. The results are presented in the table below (or Table 1 in the submitted PDF file of the global response). It is evident that in both settings of $N_B=15$ and $N_B=20$, our AR2L approach demonstrates superior performance compared to PCT and RARL across various metrics in the real-world dataset.
>
> **Table 1**: Algorithm evaluation on real-world MI Dataset
>
> |      |   -   | $N_B=15$ |   -   |   |   -   | $N_B=20$ |   -   |
> |:----:|:-----:|:--------:|:-----:|:-:|:-----:|:--------:|:-----:|
> |      | $Uti$ |   $Std$  | $Num$ |   | $Uti$ |   $Std$  | $Num$ |
> |  PCT |  48.3 |    8.5   |  16.8 |   |  48.7 |   10.1   |  16.9 |
> | RARL |  48.8 |    8.8   |  16.9 |   |  48.8 |    8.3   |  16.9 |
> | AR2L |  50.2 |    8.5   |  17.4 |   |  52.9 |    6.5   |  18.3 |
>
>
>
> Q4: Demo of packing policy
>
> Thanks for your concern regarding the demo of the packing policies. We have created a video that showcases the packing processes of the PCT policy, RARL policy, and AR2L policy in both the worst-case dynamics and the nominal dynamics. To adhere to the double-blind reviewing policy, we have submitted the video link to AC. Additionally, we have included visualized packing results in our Appendix.
>
> Q5: Additional calculations and latency of the algorithm
>
> We thank the reviewer for bringing this out. In our conclusion and limitation section, we commented that the ExactAR2L algorithm introduces additional computational complexity *in the training phase due to the mixture-dynamics attacker*. The resulting robustness is coming from the additional training of the attacker. However, in the bin packing research community and industrial applications, the computational complexity during the inference phase is more important, as we require the packing strategy to efficiently determine the location for each item. From this perspective, AR2L does not introduce additional complexity during inference compared to RARL and PCT.
>
>
> [1] Panaganti, K., Xu, Z., Kalathil, D., Ghavamzadeh, M. (2022). Robust reinforcement learning using offline data. Neurips 2022
> [2] Samir Elhedhli, Fatma Gzara, and Burak Yildiz. Three-dimensional bin packing and mixed-case palletization. INFORMS Journal on Optimization 2019.

---

### Official Review · Reviewer_HCvp · 2023-07-06

**Soundness:** 3 good
**Presentation:** 4 excellent
**Contribution:** 3 good
**Rating:** 7
**Confidence:** 5

**Summary:**

This paper investigates the online three-dimensional bin packing problem (3D-BPP) and extends the PCT algorithm by proposing an adjustable robust reinforcement learning (AR2L) framework that balances the performance of policies in average and worst-case scenarios. The paper designs a permutation-based adversary and introduces an objective function that combines expected return and worst-case return with weights. A lower bound is derived to guide policy learning. The paper presents two algorithms to implement the AR2L framework, one exact that requires training a hybrid dynamics adversary, and another approximate that samples from the original dynamics and permutation-based adversary. Experiments are conducted in both discrete and continuous settings, demonstrating the effectiveness and superiority of the AR2L framework.

**Strengths:**

1. The problem addressed in the paper is highly significant. The authors extend the state-of-the-art PCT algorithm by optimizing its worst-case performance through an adversarial approach. I also have a strong impression of the PCT work, as it intuitively and effectively solves the important online bin packing problem. It is great to see the functionality of PCT being further enhanced through this work.
2. I appreciate the organization of the paper. The authors demonstrate a strong understanding of packing and reinforcement learning, supporting each argument with data or theory. The paper is clear, understandable, and reasonable, making it enjoyable to read.
3. The proposed approach is clever. The authors introduce a novel robust reinforcement learning framework, where a permutation-based adversary is designed to generate worst-case problem instances. This is an innovative and practical method. Additionally, the paper derives a lower bound for an objective function and utilizes it to guide policy learning, providing a theoretical contribution.
4. The paper presents two algorithms to implement the AR2L framework, one exact and one approximate, catering to different scenarios and requirements.
5. The paper conducts extensive experiments, comparing the AR2L framework with multiple packing and benchmark methods, demonstrating its advantages in terms of average performance and robustness.

**Weaknesses:**

The paper does not provide guidance or suggestions on how to choose the hyperparameters α and ρ. I noticed that the optimal values of α vary when β changes in the best-performing AR2L algorithm. Perhaps this could be considered as future work.

**Questions:**

1. If I understand correctly, in Table 2 of the main text and Table 2 in the appendix, each trained policy is still tested using data generated by the attacker (as the performance of the baseline algorithms varies with different beta values, indicating a change in the tested data). So, if we remove the attacker and everyone uses the same data for testing, what would be the performance of AR2L? I am very interested in the practical implications of packing policies and would like to know if the packing policy learned through adversarial training still has advantages when tested on normal data (e.g., object sizes following a uniform distribution). This inquiry is purely out of curiosity and will not affect my scoring.
2. In the appendix, the authors mention that they failed to reproduce some PCT results. Based on my own attempt, I believe PCT is quite reliable. I suggest that the authors contact the PCT authors as they may be able to assist in successfully reproducing the results. It is possible that there are some issues related to the usage or implementation, as packing involves various complex settings that require consideration of multiple aspects, such as discrete domain, continuous domain, stability checks, and data types. The functionality of the code can be quite complex. I believe the authors of PCT will be willing to help you successfully reproduce the results, and I have sent them a message to notify them and draw their attention to possible coming inquiry emails.
3. A quick question: I would like to reproduce the results mentioned in the paper. In the code provided, what does "bal_policy" represent? Does the code refer to the code for ExactAR2L or ApproxAR2L, or is there an option to switch between the two algorithms?

**Limitations:**

The paper only considers one type of permutation-based attacker and does not explore other possible attack methods, such as adding or removing items, modifying the size or shape of items, etc.

----------------------After Rebuttal----------------------

The reviewer appreciates the author's response. However, the reviewer is still considering whether it is necessary to propose a new candidate generation method. During the rebuttal period, the reviewer downloaded the official implementation of PCT. The final test results were even higher than those reported in the PCT paper. Because the reviewer does not see a direct response in the rebuttal, holding onto this doubt, the reviewer intends to slightly lower the rating.

---

> ### Author Rebuttal · Authors · 2023-08-09
>
> We sincerely appreciate the reviewer for recognizing our contribution in developing the permutation-based attack method and the adjustable robust reinforcement learning algorithm. We are grateful for the reviewer's valuable feedback, and we would like to address their concerns as follows.
>
>
> Q1: The settings of hyperparameters
>
> Thank you for raising the concern about the hyperparameters in our AR2L framework.
> Based on observations from Table 2, the larger value of $\alpha$ (i.e., $\alpha=1$) is the best choice for ExactAR2L across different test settings. In tasks where $\beta=50, 75$, ExactAR2L with $\alpha=1$ performs the best compared to the baseline methods and ExactAR2L with other values of $\alpha$ (with a slight drop compared to ExactAR2L with $\alpha=0.7$ in the setting of $\beta=50, N_B=15$). If $\beta=100$, ExactAR2L with $\alpha=1$ can still produce competitive results compared to RARL and significantly outperforms PCT. In the task of $\beta=25$, although $\alpha=1$ is not the optimal choice for ExactAR2L, ExactAR2L with $\alpha=1$ can still outperform other baselines. In the task of $\beta=0$, ExactAR2L with $\alpha=1$ significantly outperforms RARL, and the slight drop of ExactAR2L with $\alpha=1$ compared to PCT is acceptable, as our goal is to improve the robustness while maintaining average performance at an acceptable level.
>
>
> The parameter $\rho$ is only used in ApproxAR2L. As shown in Figures 3(c) and 3(d), we chose different values of $\rho = 0.1, 0.2, 0.3, 0.4$ in different settings of $N_B$ and $\beta$. We found that $\rho=0.1$ is a trustworthy choice for ApproxAR2L. Based on the observations from Table 2 and Figures 3(c) and 3(d), we conclude that $\rho=0.1$ and $\alpha=0.5$ is the best choice for ApproxAR2L, as it outperforms its corresponding baseline (RfMDP) in almost all the tasks. In the main text, we will justify more on the choices of hyperparameters, and will also investigate a more integrated way for adjusting these parameters in robust reinforcement learning settings.
>
>
> Q2: The setting of testing data and testing on normal data
>
> Thank you for your insightful feedback regarding the experiment section. In Table 2, the column for $\beta=0$ indicates that all the algorithms were exposed to the same dataset from the nominal dynamics, while for other $\beta$ values, we construct mixture datasets by randomly selecting $\beta\%$ nominal box sequences and reordering them using the learned permutation-based attacker for each packing policy. We observed that in all tasks where $\beta=0$, ExactAR2L with $\alpha=1$ outperformed RARL, as RARL was too conservative to handle instances from the nominal dynamics. When compared to PCT, ExactAR2L with $\alpha=1$ showed a slight drop in space utilization in the $\beta=0$ tasks. This was because ExactAR2L had to consider instances from both the nominal and worst-case dynamics. If the deviation between the perturbed problem instance distribution and the nominal distribution was too large, the performance of ExactAR2L in the nominal dynamics would be degraded. However, when such deviation was constrained in an acceptable range, the training instances were diversified due to the mixture-dynamics attacker, improving the generalizability of the packing policy and resulting in better performance compared to PCT.
>
>
> Q3: Regarding the setting and implementation of PCT
>
> Thank you for expressing your concerns regarding the implementation of the baseline method PCT. After reproducing it, we found that we could only obtain good results in the discrete settings with default hyperparameter settings. However, despite trying different hyperparameter configurations, we failed to reproduce the results as reported in PCT for the continuous setting. Therefore, we decided to propose a new heuristic method, the Intersection Point heuristic, to generate candidate locations in the continuous setting. And please refer to our Appendix 1.4 for more details on the Intersection Point heuristic. This approach provided satisfactory results for us. We appreciate your suggestions and will contact the authors of PCT to reproduce their results.
>
>
> Q4: Code naming
>
> We apologize for the confusion caused by the naming conventions used in our code. To clarify, in our code, "BPP_policy" refers to the packing policy, "adv_policy" corresponds to the permutation-based attacker used for the worst-case dynamics, and "bal_policy" corresponds to the mixture-dynamics attacker. Thank you for asking about reproducing our work.
>
>
>
> Q5: Implementation of other possible attacks
>
> Thank you for expressing your interest in extending our framework. The main focus of our paper is on the robustness against permutation-based attackers. And in practice, such permutation is happening and has to be considered for improving algorithm robustness. Other types of attack methods like you suggested could be explored in future work to study the extendibility of our method.

---

> ### Author Response · Authors · 2023-08-17
>
> We appreciate the feedback provided in the review. In fact, we found that the official implementation of the baseline method PCT was just updated 3 weeks ago. Consequently, we were only able to download the older version of PCT before the Neurips deadline. When attempting to reproduce PCT following the official instructions, we did not achieve satisfactory performance in the continuous setting. As a result, we made the decision to use a new heuristic method for candidate generation in our experiments.
>
> In addition, we would like to clarify that our study specifically focuses on the robustness of the packing policy, rather than exploring alternative candidate generation methods. Therefore, to maintain a fair comparison in the continuous setting, we ensured that the same heuristic method was used for generating all candidate positions across all approaches.
>
> We appreciate the reviewer’s thorough understanding and insightful comments regarding our method. We are grateful for the valuable feedback provided, and will emphasize in the revised paper about the settings of the candidate generation method. If there are any additional concerns or questions, we are more than happy to provide further explanations and address them to the best of our abilities.

---

> > ### Comment · Reviewer_HCvp · 2023-08-18
> >
> > Thank you for your response. As I am interested in the packing problem, I hope to avoid unnecessary complexities that might confuse the community. I took the initiative to reach out to the authors of the PCT paper in hopes that they could provide assistance and update their usage documentation. However, in the authors' response to this article, I did not observe them presenting any persuasive experimental results, which is why I lowered my rating.
> >
> > Nonetheless, because I still find great value in this paper, ensuring the worst-case performance for the packing problem is meaningful. I acknowledge the authors' efforts; however, I urge them to thoroughly replicate the performance of existing methods in the revision. Based on these conditions, I am inclined to raise my recommendation to a score of 7.

---

> > > ### Author Response · Authors · 2023-08-18
> > >
> > > We sincerely appreciate the reviewer’s insightful comments and valuable feedback on our work. Your concerns and suggestions are highly valued and contribute to the packing problem research community. We acknowledge the importance of thoroughly reproducing the performance of the baseline methods using the latest official implementation, as you have recommended. In the revision, we will carefully address this concern and ensure accurate and up-to-date comparisons with the baseline methods. We are grateful for your valuable feedback, which has helped us improve the quality of our work.

---

### Official Review · Reviewer_Em3c · 2023-07-06

**Soundness:** 3 good
**Presentation:** 3 good
**Contribution:** 3 good
**Rating:** 6
**Confidence:** 2

**Summary:**

This work addresses the problem of 3D bin packing problem (3D-BPP). Specifically, it develops a permutation-based attacker and subsequently proposes an adjustable robust reinforcement learning (AR2L) framework. This allows an algorithm to consider both the average and worst-case performance with the attacker, where the packing objective is a weighted sum of expected and worst-case returns over the space utilization rate. The proposed framework is integrated with prior work to develop the exact and approximate variant of the proposed AR2L.

--------------------------
I acknowledge the author's effort in the rebuttal and have made changes to the review accordingly.

**Strengths:**

+ This work proposes to train a novel permutation-based attacker (RL-based policy) that can re-order the sequence of the observed items to reduce the space utilization of bin packaging tasks. The permutation-based attacker is used to quantitatively evaluate a given algorithm's performance under the worst-case scenario. The empirical results show new insight into the robustness of existing works.

+ This work developed an adjustable robust reinforcement learning (AR2L) approach, which learns a packing policy based on both average scenarios and worst-case scenarios. This can be achieved via a surrogate problem where an optimal policy can be identified with the maximal lower bound.

+ Two forms of implementation of the AR2L algorithm are shown in this work. The exact AR2L algorithm performs better worst-case performance under the worst-case attack with additional computation to the entire framework. On the other hand, the approximate AR2L algorithm introduces estimation errors with the cost of performance drop. For both variants, it outperforms the corresponding baseline methods.

**Weaknesses:**

- This work shows comprehensive evaluation with different combination of \alpa, \beta, and N_B parameters. While this provides valuable insights, it is unclear what would be the most optimal configuration in real-world deployment since the conditions could vary across task.

**Questions:**

- In eqn 2, should the last term be  \alpha d( P^m || P^w )?

**Limitations:**

No immediate limitations or impact raised from the novelties from this work.

---

> ### Author Rebuttal · Authors · 2023-08-09
>
> We would like to express our sincere appreciation to the reviewer for acknowledging our contribution in developing the permutation-based attack method and the adjustable robust reinforcement learning algorithm. We are grateful for the feedback provided by the reviewer, and below, we address these concerns in detail. We also provide additional experimental validations and further explanations of the settings in the global response.
>
> Q1: Configurations of parameters and variables
>
> The variables $\beta$ and $N_B$ are used to indicate and evaluate the varying difficulty levels of problem instances used in the testing phase. Therefore, $\beta$ and $N_B$ do not need to be tuned for best performance, and here we detail how $\alpha$ affects the policy's performance. Based on observations from Table 2, the larger value of $\alpha$ (i.e., $\alpha=1$) is the best choice for ExactAR2L across different test settings. In tasks where $\beta=50, 75$, ExactAR2L with $\alpha=1$ performs the best compared to the baseline methods and ExactAR2L with other values of $\alpha$ (with a slight drop compared to ExactAR2L with $\alpha=0.7$ in the setting of $\beta=50, N_B=15$). If $\beta=100$, ExactAR2L with $\alpha=1$ can still produce competitive results compared to RARL and significantly outperforms PCT. In the task of $\beta=25$, although $\alpha=1$ is not the optimal choice for ExactAR2L, ExactAR2L with $\alpha=1$ can still outperform other baselines. In the task of $\beta=0$, ExactAR2L with $\alpha=1$ significantly outperforms RARL, and the slight drop of ExactAR2L with $\alpha=1$ compared to PCT is acceptable, as our goal is to improve the robustness while maintaining average performance at an acceptable level.
>
> The parameter $\rho$ is only used in ApproxAR2L. As shown in Figures 3(c) and 3(d), we chose different values of $\rho = 0.1, 0.2, 0.3, 0.4$ in different settings of $N_B$ and $\beta$. We found that $\rho=0.1$ is a trustworthy choice for ApproxAR2L. Based on the observations from Table 2 and Figures 3(c) and 3(d), we conclude that $\rho=0.1$ and $\alpha=0.5$ is the best choice for ApproxAR2L, as it outperforms its corresponding baseline (RfMDP) in almost all the tasks.
>
>
> We would also like to note the AR2L algorithm has the practicality in real-world scenarios. We directly evaluated our model on the Mixed-item Dataset (MI Dataset) which follows the generation scheme proposed by Elhedhli et al. [1] for the realistic 3D-BPP instance generator. MI dataset has 10 thousand items, with 4668 species, and occurrences vary from 1 to 15. The pallet dimensions is set to the size often used in practice: L = 120, W = 100, and H = 100. The results are presented in the table below (or Table 1 in the submitted PDF file of the 'global' response). It is evident that in both settings of $N_B=15$ and $N_B=20$, our AR2L approach demonstrates superior performance compared to PCT and RARL across various metrics in the real-world dataset.
>
>
> **Table 1**: Algorithm evaluation on real-world MI Dataset
> |      |   -   | $N_B=15$ |   -   |   |   -   | $N_B=20$ |   -   |
> |:----:|:-----:|:--------:|:-----:|:-:|:-----:|:--------:|:-----:|
> |      | $Uti$ |   $Std$  | $Num$ |   | $Uti$ |   $Std$  | $Num$ |
> |  PCT |  48.3 |    8.5   |  16.8 |   |  48.7 |   10.1   |  16.9 |
> | RARL |  48.8 |    8.8   |  16.9 |   |  48.8 |    8.3   |  16.9 |
> | AR2L |  50.2 |    8.5   |  17.4 |   |  52.9 |    6.5   |  18.3 |
>
>
> Q2: Typo in Equation
>
> We apologize for the typo in Equation (2), where the second term on RHS should be $(d(P^{m}||P^{o}) + \alpha d(P^{m}||P^{w}))$. We will correct the references to the equations in our paper. We thank the reviewer for bringing up these concerns and questions. We hope our explanations have addressed these concerns.
>
> [1] Samir Elhedhli, Fatma Gzara, and Burak Yildiz. Three-dimensional bin packing and mixed-case palletization. INFORMS Journal on Optimization 2019.

---

> > ### Comment · Reviewer_Em3c · 2023-08-16
> >
> > I acknowledge that I have read the rebuttal and the authors has responded to my concerns.

---

### Official Review · Reviewer_fVdw · 2023-07-25

**Soundness:** 3 good
**Presentation:** 2 fair
**Contribution:** 2 fair
**Rating:** 4
**Confidence:** 4

**Summary:**

The paper is in the category of work that uses reinforcement learning for combinatorial optimization problems. More specifically, it focuses on the online version of the 3d-bin-packing problem (3D-BPP), and proposes a robust RL solution to address the uncertainty that arises from the permutation of an adversarial nature. At the core of their solution, Adjustable Robust Reinforcement Learning (AR2L), is 1) a new attacker that is based on permutation of the observable sequence of the items, 2) a new objective function that adjusts weights between nominal performance and worst-case performance, and 3) theories and algorithms that solve the new optimization.

**Strengths:**

+ RL for combinatorial optimization is an interesting and trending topic. It can potentially draw interest from the RL/OR community.

+ Robustness in 3D-BPP is a practical and important issue.

+ Permutation-based attacker makes sense.

+ The adjustable objective makes sense.

+ The evaluation of robustness for existing methods is applausible.

+ Connections of AR2L to RARA and RfMDP are interesting.

**Weaknesses:**

- The attacker capacity seems questionable. Because of the following reasons, the problem setting seems rather impractical. These make the formulation, theorems, and evaluation somewhat trivial.
1) If you formulate the nature as an adversary, then the nature should have control over the permutation of the entire sequence of items that are not presented to the packing policy, not just the observable ones.
2) Also, it seems that the evaluation assumes the packing policy only observes one item. This is a bit weired because in this case the attacker does not have any capacity.
3) Moreover, the attacker's policy is to change only one item to the most preceding item, but the attacker should be able to change the entire permutation. This is really how the attacker makes the training robust, and evaluations should be conducted against such an attacker.
4) Finally, the setting that the sequence of the observable items can be changed is also weired. This is because the packing agent/policy has already seen the set of observable items, you cannot suddenly change them without the notice from the packing agent.

- The paper, along with its formulation, seems to be too much restricted to robustness to permutation-based attack in 3D-BPP. Do the theories/results generalize to other settings? There is a lack of discussion from that aspect. This makes the technical contribution rather limited. The paper would be stronger if it can shed light on how the algorithms are generally applicable to more combinatorial optimization problems with uncertainty.

- Theorem 1: $\gamma$ is not explained -- is it the discount factor? Isn't is assumed to be 1 in the Preliminaries section? If so, the denominator is 0. At least $1-\gamma$ is a very small value, making the second term on the RHS of Equation (2) dominating the entire RHS. This significantly restricts the meaningfulness of Theorem 1 (a major theoretical contribution). It seems that the objective in Equation (3) makes sense by itself, but without Theorem 1 this is only a heuristic. Also, why the coefficients are thrown out in Equation (2) and turned into Equation (3) was not explained.

- The empirical results are not convincing
1) it looks like that the average performance of AR2L is outperformed by the baselines (esp. RARL) in many cases (esp. in the most challenging setting where $\beta=100$. It is surprising that PCT is not the best even for the nominal setting ($\beta=0$). These have not been discussed and explained.
2) Also, although the paper is about robustness, it primarily focuses only on average performance (Uti), discussions of Std are limited even thought the results are shown.
3) I was expecting to see some guidelines/lessons-learned from what values I should take for the hyperparameters $\alpha$ and $\rho$, but did not see. If I were to use the algorithms, how do I choose them? It seems that if I choose the wrong ones, I might end up with a solution that is worse than the baselines. The mysterious performances regarding hyperparas are very un-convincing.

Minor issues:
* Refereces to equations should be Equation (1), instead of 1. See e.g., Line 205.
* A typo at the end of Equation (2). $P^0$ -> $P^w$.
*

**Questions:**

See questions in the weakness part.

---

> ### Author Rebuttal · Authors · 2023-08-09
>
> Thank you for taking the time to review our paper. We appreciate your valuable feedback. However, we would like to clarify that there may be some misunderstandings regarding our settings. We would like to take this opportunity to address your concerns. Some shared concerns about AR2L's empirical performance is additionally addressed in global author rebuttal with additional results to broader RL tasks.
>
> Q1: The settings of attacker
>
> **Adversary setup**
> We thank the reviewer for asking about the attacker capability. In the online bin packing, we assume two crucial components can control the entire item permutation, namely the item distribution and the permutation of the observable items. Since these two components are included in the environment for the packing policy, the attacker aims to change the environment parameter/setting by permuting the observable items. This setting aligns with RARL, which investigates the robustness against environment parameter changes.
>
> Most importantly, we want to emphasize that the permutation-based attacker's objective is to identify challenging instances by solely reordering the item sequence without selectively favoring specific types of items. But upon analyzing Figure 1 in Appendix, we observed that as the number of observable items increases, the attacker appears to prefer smaller items to make the instance harder. So if we do not limit the attacker's capabilities, it may trickily select certain types of items to construct harder instances, which goes against our intended goal.
>
> **Attacker capacity**
> In section 5.1, it is important to note that the evaluated packing methods except PCT, can only observe the first item at each step. To ensure a fair comparison for the robustness evaluation, we set the number of observable items for packing policies to 1. To make the attacker effectively perturb the packing policy, *we allow the attacker to observe more items (i.e. $N_B=5, 10, 15, 20$)*, which is also aligned to changing the environment parameter/setting. Our evaluation ensures a fair comparison for the robustness studies, while also recognizing the attacker's need for more information about the item sequence to effectively perturb the packing policy in this scenario.
>
> In section 5.2, we use PCT as the packing policy in all the algorithms, which allows the *packing policy to observe multiple items at each step*. Therefore, we set the number of observed items for both the attacker and the packing policy to be the same. It still provides a fair comparison for the robustness evaluation. In addition, although the attacker does not obtain more information compared to the packing policy, they can still effectively perturb the packing policy.
>
>
> **Attacker's permutation**
> We thank the reviewer for bringing the setup issue out. In our designed attacker, the entire permutation is progressively changed as the time step increases. At each step, the attacker chooses to move one item from the observed items to the most preceding position. Since the attacker aims to choose the item with the highest potential to decrease the space utilization, *the entire permutation is actually altered throughout the attack*. By contrast, in the unperturbed item sequence, each item is randomly sampled and does not reduce the space utilization adversarially. Thus, the packing policy can learn to improve its robustness with the presence of item sequences progressively generated by the attacker.
>
>
> **Sequence of observed items**
> As we mentioned earlier, the environment consists of both the item distribution and the permutation of observable items. The attacker first permutes the item sequence to change the environment, aligned to RARL. Then the packing policy packs the first item in the reordered item sequence.
>
>
> Based on these justifications, our setting of the permutation-based attacker with limited observable items aligns with practical considerations and our intended objectives. This ensures that our formulation and theorems are reasonable. By conducting a fair comparison in our evaluation, we verify the value and effectiveness of our work. It is important to note that our study provides the first comprehensive analysis of adjustable robustness by attacker training and integration, which is a nontrivial setup.
>
>
>
> Q2: The applicability of AR2L.
> Theorem 1 is general and derived from the standard MDP, and the adjustable Bellman operator in Theorem 2, is also a general operator like the robust Bellman operator. We are confident that our framework can be extended to other combinatorial optimization problems with uncertainty and applied to various RL problems beyond the scope of this study (please see global response).
>
> Q3: Theorem 1's parameter.
> We thank the reviewer for pointing this out. $\gamma$ is the discount factor in Theorem 1. And we want to emphasize that Theorem 1 is a general theorem. In our paper, it provides insight and guidance for estimating the lower bound of the general objective in Equation 1. Specifically, Theorem 1 suggests that to increase the lower bound of our objective, it is an option to increase $\eta(\pi, P^{m})$ and $-(d(P^{m}||P^{o}) + \alpha d(P^{m}||P^{w}))$. Based on this analysis, we propose the new objective in Equation 3. We believe that this operation and objective transformation, guided by the insight from Theorem 1, is reasonable and commonly used in the field of RL. For example, in TRPO, the use of a similar coefficient can result in very small update step sizes, which is why they use the trust region constraint and discard the coefficient for practical implementation.
>
>
> Q4: Performance evaluation.
> We apologize that due to the limited space we include the discussions about performance comparisons, metrics of $Std$ and $Num$, and the selection of hyperparameters to the global response. Please refer to the global response for details.
>
>
> ### Minor Issues
> We apologize for typos and refereces in equations and draft, and will revise them accordingly.

---

> > ### Comment · Reviewer_fVdw · 2023-08-16
> >
> > I thank thre authors for the responses. However, I am not entirely convinced by the issues around the attacker's capacity, the empirical evaluation, and the hyperparameters.
> >
> > In terms of the attacker's capacity, e.g., why doesn't the attacker permute the entire sequence of observable items instead of just putting one item ahead of the others? Although eventually the attacker will change the order of the entire sequence via multiple steps, this limits the power of the attacker. Also, why doesn't the attacker have the capacity of changing the item distribution (like Kong et al.)?
> >
> > In terms of the empirical evaluation. E.g., it is not convincing why PCT is outperformed by AR2L even when $\beta=0$. Does this imply that the baselines are weak in the first place? Also, the std of AR2L(1) is larger than RARL in many cases -- this is counter-intuitive since RARL focuses on the worst-case, it should have higher stds.
> >
> > In terms of the hyperparas, how about $\alpha>1$? $\alpha=1$ is the largest value being tested, but it seems that larger $\alpha$'s may yield even better performances. But this may indicate that the "robustness" does not matter that much and the algorithm reduces to maximizing the nomimal objective.

---

> > > ### Author Response · Authors · 2023-08-17
> > >
> > > Thank you for your response. We sincerely appreciate your valuable feedback. We would like to provide further clarification and address these concerns.
> > >
> > > Q1: The settings of attacker
> > >
> > > **Moving one item**
> > >
> > > From the perspective of the practical setting of online BPP, the packing policy can observe multiple items, but it can only pack the first item into the container. Let's consider a scenario where the attacker permutes the entire sequence of observable items at timestep $t$. In this case, regardless of the permutation applied to the remaining items at timestep $t$, at timestep $t+1$, the remaining items will combine with a new incoming item to form a new sequence that will be further permuted. Therefore, any permutation of the remaining items at timestep $t$ will be ignored and subsequently permuted with the new incoming items.
> > >
> > > From the perspective of the influence on the packing policy, let's consider a simplified value function $V(C_{t}, B_{t}) = r_t + V(C_{t+1}, B_{t+1})$, where $C_{t}$ and $B_{t}$ represent the bin configuration and the permuted observable item sequence, and $r_t$ is the reward. It tells us that the value of the packing policy at timestep $t$ depends on both the reward $r_t$ and the value function $V(C_{t+1}, B_{t+1})$. And $r_t$ and $C_{t+1}$ are influenced by the first item in $B_{t}$. The remaining items in $B_{t}$ will combine with a new item to form a new sequence, which will undergo further permutations as $B_{t+1}$. Consequently, the permutation of the remaining items at timestep $t$ will be disregarded in the attack process.
> > >
> > >
> > > From the perspective of the implementation, if we allow the attacker to permute the entire sequence of observable items, the action space will grow exponentially as the number of observable items increases. This large search space may have a detrimental effect on the convergence of the RL-based attacker.
> > >
> > >
> > > **Item distribution**
> > >
> > > In our paper, we specifically focus on the impact of item permutation on the packing policy, rather than the item distribution. To ensure that our study remains focused on the randomness introduced by item permutation, we design the permutation-based attacker, which allows us to avoid any influence stemming from changes in the item distribution. Furthermore, as mentioned in PCT, larger items can simplify the scenario, while smaller items can trickily make the task more challenging. Therefore, we aim to prevent the attacker from simply learning to selectively favor certain types of items to create challenging instances. Instead, we want the attacker to genuinely learn how to permute the item sequence in order to identify and create challenging scenarios. Hence, we have designed an attacker that is unable to alter the item distribution in order to maintain the integrity of our study.
> > >
> > >
> > >
> > > Q2: The evaluation
> > >
> > > **The performance of $\beta=0$**
> > >
> > > We would like to emphasize that there are inherently challenging instances present in the nominal dynamics ($\beta=0$). Therefore, one approach to improving performance when $\beta=0$ is to enhance the performance specifically on those challenging instances while maintaining performance on others. When using a smaller value of $N_B$, the distribution of training data (mixture instances) and testing data ($\beta=0$) exhibits less deviation. This enables the policy to effectively handle both challenging instances and moderate instances, thereby improving generalization. However, when $N_B=20$, the substantial deviation between the distributions can negatively impact performance when testing on $\beta=0$. And we have confidence in the strength of the baseline methods chosen due to the fair experimental comparison settings.
> > >
> > > **Standard Deviation**
> > >
> > > We would like to clarify that in a high uncertainty environment, such as online 3D BPP, robustness specifically refers to the ability of a policy to consistently perform well despite the presence of uncertainties. It is intuitive that the RARL policy will consistently exhibit a conservative behavior, regardless of the problem instances provided. This conservative behavior ensures a consistent performance and reduces the variance of the policy. On the other hand, the AR2L(1) policy demonstrates a less conservative behavior, resulting in relatively larger variance.
> > >
> > >
> > > Q3: The hyperparameter
> > >
> > > We have the flexibility to set $\alpha > 1$. However, based on observations from our empirical study, we found that larger $\alpha$ can actually lead to a degradation in performance in the nominal dynamics. This suggests that AR2L with larger $\alpha$ tends to overprioritize worst-case scenarios, similar to previous robust methods. Given that our objective is to achieve a desired balance between performance in the nominal and worst-case dynamics, it is not recommended to set $\alpha > 1$. Doing so may result in an excessive focus on worst-case performance, compromising the overall performance in the nominal dynamics.

---

> > > > ### Comment · Reviewer_fVdw · 2023-08-18
> > > >
> > > > I aprpeciate the authors' clarification. I am now more convinced by the attacker setting. But I am still not convinced by the evaluation.
> > > >
> > > > - The performance when $\beta=0$. Intuitively, isn't a regular RL policy, or AR2L(0) the best option?
> > > >
> > > > - Standard deviation -- the problem is that the performances are inconsistent here. It looks like sometimes the std of RARL is smaller than AR2L(1), but sometimes it is larger (e.g., when $N_B=10, \beta=0$ or $100$).
> > > >
> > > > - From the table, it does not seem so. It seems that there is a trend of improving for the average performance when you increase $\alpha$. Also, this is somewhat counter-intuitive -- why $\alpha=1$ yields better average performance than, e.g., when $\alpha=0.5$ or $\alpha=0$? Isn't smaller $\alpha$'s more focused on average performance?

---

> > > > > ### Author Response · Authors · 2023-08-19
> > > > >
> > > > > We express our sincere gratitude for the valuable comments provided on our work, and we are genuinely appreciative of your understanding and acceptance of our attacker settings. We are more than willing to provide further clarification on the evaluation part to address the concerns you raised.
> > > > >
> > > > > Q1: The performance when $\beta=0$.
> > > > >
> > > > > Intuitively, when testing with $\beta=0$, the regular RL policy appears to be a better option compared to AR2L with a small value of $\alpha$ (e.g., AR2L(0.3)). However, upon analyzing the empirical results, we observed that by incorporating a small quantity of challenging problem instances into the training setting of the nominal dynamics, the overall performance can actually be improved. This finding suggests that when we introduce a few challenging instances during the training process, the performance on moderate instances from the nominal dynamics ($\beta=0$) remains unaffected. This prevents an overemphasis on worst-case performance while allowing the packing policy to learn from more challenging instances. Consequently, the policy's ability to handle these inherently difficult instances from the nominal dynamics can be enhanced.
> > > > >
> > > > > We consider this observation to be an interesting finding, as it highlights the potential benefits of including a small number of challenging instances during training, leading to improved generalization. We believe that this finding provides valuable insights to the field of the packing problem and sheds light on the effectiveness of incorporating challenging instances in the training process for more general combinatorial and sequential problems.
> > > > >
> > > > >
> > > > >
> > > > > Q2: Standard deviation.
> > > > >
> > > > > Based on the observations from the empirical studies presented in Table 2 of the main text, we found that in 17 out of 20 tasks, AR2L(1) exhibits a larger standard deviation compared to RARL. Conversely, in the remaining 3 tasks, AR2L(1) shows a smaller standard deviation. We believe these results demonstrate a consistent trend. In the majority of tasks, AR2L(1) exhibits a larger standard deviation, indicating its ability to consider instances from both the nominal and worst-case dynamics. It is worth noting that even in previous robust methods, such as CPPO and RARL, their standard deviations sometimes exceed some of their respective baselines, while in other cases, their standard deviations are smaller. We believe that the consistent trend observed in the majority of tasks provides an explanation that is in line with our expectations. And we believe AR2L's formulation to efficiently adjust robustness weights is valuable compared to the relatively conservative RARL setting.
> > > > >
> > > > >
> > > > >
> > > > > Q3.
> > > > >
> > > > > To provide a thorough explanation for the observations in Table 2, let us start with the scenario where $N_B=5$. In this scenario, the attacker's ability is limited to permuting only 5 observable items. As a result, the distribution deviation between the training data (mixture data) and the testing data ($\beta=0$) is relatively small. This means that it is acceptable to increase the value of $\alpha$ and introduce more challenging instances into the training settings of the nominal dynamics. By doing so, the generalization of the model can be improved while avoiding an excessive focus on worst-case performance.
> > > > >
> > > > > However, as the number of observable items ($N_B$) increases, the distribution deviation between the training data (mixture data) and the testing data ($\beta=0$) also increases due to the attacker's improved capability. In such cases, it becomes less desirable to significantly increase the value of $\alpha$, as doing so would introduce a large number of challenging instances into the training set. This can lead to a substantial deviation between the training set and the testing set. As a result, in the cases of $N_B=10$ and $N_B=15$ under $\beta=0$, AR2L tends to favor $\alpha=0.5$.
> > > > >
> > > > > Furthermore, in the scenario where $N_B=20$, AR2L cannot outperform PCT due to the larger distribution deviation caused by the increased number of observable items of the attacker. In summary, when the attacker's capability is limited, a higher value of $\alpha$ can be beneficial. However, as the attacker's capability increases, a lower value of $\alpha$ is more suitable. Hope this address the reviewer's comment on selection of $\alpha$.
> > > > >
> > > > >
> > > > > Thank you for your response again. We genuinely appreciate your valuable feedback and the opportunity to provide further clarification on any concerns you may have. Your feedback is crucial to improving the quality and comprehensibility of our research. If there are any additional concerns or questions, we are more than happy to provide further explanations and address them to the best of our abilities.

---

> > > > > > ### Comment · Reviewer_fVdw · 2023-08-21
> > > > > >
> > > > > > Thanks for your response. Unfortunately I was not entirely convinced about the empirical evaluations. It appears to me that there is a lack of comprehensive understanding of what results were obtained and why they were obtained -- the authors seem to have a better understanding with the push of questions, but it is questionable that within a short time that these phenomena can be fully understood and well explained.
> > > > > >
> > > > > > But like I said, I am more convinced by the attacker model after the authors' responses. So I am willing to increase my rating from 3 to 4.

---

> > > > > > > ### Author Response · Authors · 2023-08-21
> > > > > > >
> > > > > > > We sincerely appreciate your comments and feedback on our work. In the revised version, we will focus on discussing the empirical results in more detail to demonstrate the significance and rationality of our findings. We are grateful for your contribution in the rebuttal process, as it has helped us improve the quality of our paper.

---

### Author Rebuttal · Authors · 2023-08-09

Dear Reviewers:

We appreciate your valuable comments and have made clarifications to all of your questions and concerns in our response. Below are some shared concerns among reviewers.

Q1: Practicability and generalizability of AR2L

To validate the practicality of AR2L in real-world scenarios, we directly evaluated our model on the Mixed-item Dataset which generates the realistic 3D-BPP instance. To assess the generalizability of AR2L, we applied AR2L to the CartPole environment. The results are presented in the submitted PDF file of the global response.
In addition, to adhere to the double-blind reviewing policy, we have submitted the link of the video demo of the packing process to AC.

Q2: Contributions of our work

In our work, we address the challenges posed by the randomness in the permutation of item sequences widely existed in online bin packing problem (BPP). To tackle this issue, we introduce the permutation-based attacker with limited capabilities. This attacker aligns with practical considerations and our research goals. To enhance the robustness of the packing policy while maintaining performance in nominal cases, we propose the AR2L algorithm based on the general theorems we derived. AR2L avoids overprioritization while also considering robustness. Our approach not only solves the BPP but also can extend to other RL problems of a more general nature.



Q3: Performance under $\beta=0$ and $\beta=100$

The AR2L policy is trained on instances from both the worst-case and nominal dynamics. When evaluating the scenario of $\beta=100$, due to the deviation between the data distributions used for training and testing, AR2L may not consistently outperform RARL. However, the objective of our paper is to *strike a balance between the policy's performance in average and worst-case environments*. As such, for $\beta=0, 25, 50, 75$, AR2L outperforms RARL, demonstrating its superiority in achieving such balancing goal.

The PCT policy is trained on instances from nominal dynamics, but it is important to note that random item permutation in the nominal dynamics can naturally produce some challenging instances. AR2L can diversify the training data by providing more challenging instances with various patterns. If instances are only generated from the nominal dynamics, challenging instances may be overwhelmed by moderate instances. By training on the diversified instances from AR2L, the policy can improve generalization, and can effectively handle both moderate and challenging instances from the nominal dynamics ($\beta=0$). However, we also recognize that larger $\alpha$ and $N_B$ may cause a large deviation between the data distributions. But AR2L still demonstrates its superiority in the cases of $\beta=25, 50, 75, 100$. It is important to keep in mind the "No Free Lunch Rule".


Q4: Discussions about $Std$ and $Num$.

As shown in Table 2, ExactAR2L demonstrates its superiority over PCT with smaller Std in 17 tasks, while producing a slightly larger Std in 3 tasks. Thus, ExactAR2L can indeed improve the robustness. Additionally, we observe when $N_B=5, 10$, ExactAR2L tends to choose $\alpha=0.7$ for smaller Std, and when $N_B=15, 20$, ExactAR2L favors $\alpha=1$. Since ExactAR2L is trained on both nominal and worst-case dynamics, while RARL is trained only on the worst-case dynamics, the ExactAR2L policy is less conservative than the RARL policy. While the conservativeness may result in smaller Std in most tasks, it produces worse results of Uti under nominal dynamics. It is worth noting that the value of Std from ExactAR2L is the closest to that of RARL. This observation shows ExactAR2L can trade off between conservative and risky behavior, as the Std from ExactAR2L is between that of RARL and PCT. Similarly, ApproxAR2L is less conservative than RfMDP, which causes ApproxAR2L cannot achieve a smaller Std in all tasks.

We use ExactAR2L(1) to denote ExactAR2L with $\alpha=1$. The ExactAR2L algorithm can pack more items in 17 tasks compared to PCT, and shows a slight drop in 3 tasks, where the average drop is 0.2. We found that to pack more items, ExactAR2L consistently favors $\alpha=1$ across various tasks. Compared to RARL, ExactAR2L(1) can pack at least the same number of items in 16 tasks. Thus ExactAR2L(1) can produce competitive results compared to RARL and PCT in terms of Num. Compared to the baseline method RfMDP, ApproxAR2L(0.5) can pack more items in 16 tasks, and shows a slight drop in only 4 tasks, where the average drop is 0.25. In the revised paper, we would take the reviewer's suggestions and give more detailed and comprehensive discussions.


Q5: Hyperparameter selection

$\beta$ and $N_B$ are used to indicate the varying difficulty levels of data used in the evaluation. Thus, they do not need to be tuned for best performance. Based on observations from Table 2, $\alpha=1$ is the best choice for ExactAR2L across different test settings. When $\beta=50, 75$, ExactAR2L(1) performs the best compared to baselines and ExactAR2L with other values of $\alpha$ (with a slight drop compared to ExactAR2L(0.7) when $\beta=50, N_B=15$). If $\beta=100$, ExactAR2L(1) can still produce competitive results compared to RARL and significantly outperforms PCT. When $\beta=25$, although $\alpha=1$ is not the optimal choice, ExactAR2L(1) can still outperform other baselines. When $\beta=0$, ExactAR2L(1) significantly outperforms RARL, and the slight drop compared to PCT is acceptable, as our goal is to improve the robustness while maintaining average performance at an acceptable level.

$\rho$ is only used in ApproxAR2L. As shown in Figures 3(c), 3(d), we chose different values of $\rho$ in different settings. We found that $\rho=0.1$ is a trustworthy choice for ApproxAR2L. Based on the observations from Table 2 and Figures 3(c), 3(d), we conclude that $\rho=0.1$ and $\alpha=0.5$ are the best choice for ApproxAR2L, as it outperforms its corresponding baseline in almost all the tasks.

---

### Decision · Program_Chairs · 2023-09-21

**Decision:**

Accept (poster)

**Comment:**

The paper has received mixed reviews. On the positive side, the paper has been commended for presenting a clever approach validated on an extensive set of experiments, which were confirmed by one reviewer. On the more skeptic side there have been inquiries into experimental concerns, regarding some of the choices in the adversary as well as comprehensiveness.
The AC follows reviewer's HCvp assessment. The authors' rebuttal has been acknowledged. The AC urges the authors to take into account fVdw concerns and address them in the final publication.